# Inhibition of urease-mediated ammonia production by 2-octynohydroxamic acid in hepatic encephalopathy

Diana Evstafeva[1], Filip Ilievski [1], Yinyin Bao [1], Zhi Luo [1], Boris Abramovic[1], Sunghyun Kang[1], Christian Steuer [1], Elita Montanari[1], Tommaso Casalini[2], Dunja Simicic[3,4], Dario Sessa[5], Stefanita-Octavian Mitrea[3,4], Katarzyna Pierzchala[3,4], Cristina Cudalbu[3,4], Chelsie E. Armbruster [6] & Jean-Christophe Leroux [1] ✉

Hepatic encephalopathy is a neuropsychiatric complication of liver disease which is partly associated with elevated ammonemia. Urea hydrolysis by urease-producing bacteria in the colon is often mentioned as one of the main routes of ammonia production in the body, yet research on treatments targeting bacterial ureases in hepatic encephalopathy is limited. Herein we report a hydroxamate-based urease inhibitor, 2-octynohydroxamic acid, exhibiting improved in vitro potency compared to hydroxamic acids that were previously investigated for hepatic encephalopathy. 2-octynohydroxamic acid shows low cytotoxic and mutagenic potential within a micromolar concentration range as well as reduces ammonemia in rodent models of liver disease. Furthermore, 2-octynohydroxamic acid treatment decreases cerebellar glutamine, a product of ammonia metabolism, in male bile duct ligated rats. A prototype colonic formulation enables reduced systemic exposure to 2-octynohydroxamic acid in male dogs. Overall, this work suggests that urease inhibitors delivered to the colon by means of colonic formulations represent a prospective approach for the treatment of hepatic encephalopathy.

Hepatic encephalopathy (HE) is a neuropsychiatric disturbance caused by liver disease and/or portosystemic shunting[1]. HE manifestations range from subclinical changes to more serious health consequences such as coma[1]. They are determined by the severity and etiology of underlying liver disease as well as comorbidities (e.g. diabetes, kidney failure)[2]. HE is frequent in cirrhotic patients and often leads to hospitalizations and repeated readmission, thereby accounting for substantial healthcare utilization and costs[3]. Therefore, with the increasing incidence of chronic liver disease[4], effective therapies for HE remain an unmet medical need.

Although various factors are involved in the pathogenesis of HE, ammonia is assumed to play a central role in this process, and remains a prime target for current therapies[2,5]. Ammonia mainly originates from protein degradation, deamination of amino acids and urea hydrolysis by urease-producing bacteria in the gastrointestinal (GI) tract[2]. Its concentration in the systemic circulation is primarily regulated by hepatic metabolism[6]. Upon absorption, gut-derived ammonia is detoxified via the urea cycle in periportal hepatocytes and is excreted as urea through the kidneys. Furthermore, glutamine synthetase activity in various organs, such as the liver, muscles, and brain

[1]Institute of Pharmaceutical Sciences, Department of Chemistry and Applied Biosciences, ETH Zurich, Zurich, Switzerland. [2]Institute for Chemical and Bioengineering, Department of Chemistry and Applied Biosciences, ETH Zurich, Zurich, Switzerland. [3]CIBM Center for Biomedical Imaging, Lausanne, Switzerland. [4]Animal Imaging and Technology, EPFL, Lausanne, Switzerland. [5]Swiss Pediatric Liver Center, Department of Pediatrics, Gynecology and Obstetrics, University Hospitals Geneva and University of Geneva, Geneva, Switzerland. [6]Department of Microbiology and Immunology, Jacobs School of Medicine and Biomedical Sciences, State University of New York at Buffalo, Buffalo, NY, USA. ✉e-mail: jleroux@ethz.ch

enables ammonia detoxification through the synthesis of glutamine (Gln)[6]. In liver disease, impaired hepatic metabolism leads to elevated systemic ammonia concentrations (i.e. hyperammonemia), which is deleterious for the brain[7]. In the brain, ammonia is normally utilized by astrocytes to synthesize Gln[8]. However, under hyperammonemic conditions, ammonia detoxification leads to Gln accumulation in astrocytes and thus to osmotic imbalance, oxidative stress, disruption of cell metabolism and neurotransmission as well as other alterations causing various neuropsychiatric symptoms[2,9,10].

Current HE treatments are centered around the regulation of ammonia production, absorption or elimination from the systemic circulation[2]. Non-absorbable disaccharides (e.g. lactulose or lactitol) and poorly absorbed antibiotics (e.g. rifaximin) are the foundation of HE treatment. Lactulose is a laxative that reduces ammonia absorption by accelerating intestinal transit and subsequent ammonia passage from the colon[2]. It is, however, associated with unpleasant GI symptoms impairing patients' compliance[11]. Rifaximin is a broad-spectrum antibiotic that exhibits minimal systemic absorption, and therefore exclusively depletes gut bacteria reducing ammonia production[12]. Recent studies suggest that besides its antimicrobial activity, rifaximin might alter bacterial metabolism as well as exhibit anti-inflammatory effects[13]. Due to its low bioavailability, rifaximin is generally safe and well-tolerated[14]. The long-term use of rifaximin, however, raises concerns over the development of antibiotic resistance, although former work seems to indicate that the risk is low[15,16].

Urease-producing bacteria are historically perceived as one of the key contributors to intestinal ammonia[2]. In healthy individuals, approximately a quarter[17] of urea produced in the liver is broken down by bacterial ureases in the GI tract to ammonia and carbamate, which spontaneously decomposes to a second molecule of ammonia and carbon dioxide[18]. The catalytic mechanism of urease as well as enzyme's structural features have been extensively addressed in prior literature[19–21]. In fact, the architecture of the ureases' active site seems to be conserved across organisms[19–21]. In the active site, ureases contain two nickel ions coordinated by carbamylated lysine[19–21]. One nickel ion is additionally coordinated by two histidine residues, while the other one is coordinated by other two histidine and one aspartate residues[19–21]. Furthermore, a hydroxide ion bridging nickel ions combined with three terminal water molecules create a H-bonded water tetrahedral cluster occupying the active site cavity[19–21]. Besides the residues of the active site, the catalytic function of ureases is influenced by the mobile flap which is composed of a flexible helix−turn−helix motif, and is believed to serve as a gate for the substrate[19]. The presence of nickel ions in the active site renders ureases susceptible to nickel-binding compounds such as hydroxamic acid (HAs). Acetohydroxamic acid (AHA) is the most studied derivative within this group of compounds and the sole approved urease inhibitor indicated as adjunctive therapy for chronic urea-splitting urinary infection. The binding mode of AHA was elucidated in four X-ray crystal structures of ureases complexed with AHA (PDB ID: 4H9M[22], 4UBP[23,24], 1E9Y[25,26] and 1FWE[27,28]) and was found to be similar across them (the active site and binding pose are shown in Supplementary Fig. 1). In fact, AHA inhibits ureases by interacting with nickel ions. Specifically, the hydroxyl oxygen of AHA bridges both nickel ions, while the carbonyl oxygen interacts with a single nickel ion leading to the substitution of hydroxide and water molecules within the active site[20,23,27]. The binding is further enhanced through the formation of hydrogen bonds between AHA and urease residues near the active site[20]. Pearson et al.[27] described a hydrogen bond between the AHA's carbonyl oxygen and the Nε atom of a neighboring histidine (His219), as well as between the nitrogen atom of AHA and the carbonyl oxygen of alanine (Ala363) in *Klebsiella aerogenes* urease (PDB ID: 1FWE)[27,28]. The presence of another hydrogen bond involving an NH group of AHA and a side chain carboxyl group of an aspartate residue (Asp363) in the active site was suggested for *Bacillus pasteurii* urease (PDB ID:

4UBP)[23,24]. Similar to the urease-AHA complex structures mentioned earlier, a recent Cryo-EM structure of *Helicobacter pylori* urease with another HA-based inhibitor, 2-{[1-(3,5-dimethylphenyl)−1H-imidazol-2-yl]sulfanyl}-N-hydroxyacetamide (SHA, PDB ID: 6ZJA[29,30]), demonstrated the coordination of nickel ions by the inhibitor in the active site. Furthermore, van der Waals interactions between the side chain of SHA and urease's mobile flap were reported[29].

Owing to their anti-ureolytic activity, HAs have been previously investigated as the treatment for HE[31–35]. AHA was the first derivative of the class to be tested in patients with liver disease[31,32]. In a single-patient study, Fishbein et al.[31] showed for the first time a transient decline of the blood ammonia concentrations following administration of AHA (32 and 48 mg/kg). In the later study by Summerskill et al.[32], where a solution of AHA (20−60 mg/kg) was administered via the intragastric tube to the late-stage liver disease patients, one out of five patients showed a significant clinical improvement and decreased blood ammonia levels. Besides AHA, other more potent urease inhibitors, such as octanohydroxamic acid (OHA)[33,34] and nicotinohydroxamic acid[35], were subsequently studied as potential drugs for treating HE. Although these HAs decreased blood ammonia levels in some patients, none of them were further clinically developed, possibly due to insufficient activity, potential toxicity and low concentration at the target site (i.e. colon).

In the present study, we hypothesized that a potent HA-based urease inhibitor delivered to the colon could constitute a promising treatment for HE. A series of aliphatic saturated and unsaturated HAs was synthesized and screened for the ability to inhibit ureases in rat caecal content. Among them, 2-octynohydroxamic acid (2-octynoHA), was identified as a promising lead candidate and further characterized in terms of stability, cytotoxicity, mutagenicity and permeability. The pharmacokinetics (PK) of 2-octynoHA was investigated in dogs following the administration of a prototype colonic formulation, while the compound's in vivo efficacy was assessed in rodent models of hyperammonemia.

## Results and Discussion
### Design and synthesis of HAs
Inspired by early work from Kobashi et al.[36,37] which reported the in vitro anti-ureolytic activity of various aliphatic saturated HAs, we first synthesized and tested a series of saturated HAs with alkyl chain lengths ranging from 7 to 12 carbon atoms: heptanohydroxamic acid (HHA), nonanohydroxamic acid (NHA), decanohydroxamic acid (DHA), laurohydroxamic acid (LHA). AHA and OHA (2 and 8 carbons alkyl chains, respectively) were commercially available, and therefore, were included to the screening set without additional purification. Considering the superior potency of OHA over other saturated aliphatic HAs, which was initially reported by Kobashi et al.[36,37] and later confirmed by our experiments (see below), a series of unsaturated derivatives of OHA with double or triple bonds at positions 2, 3 or 7 was produced in order to investigate the impact of unsaturation on the anti-ureolytic activity of HAs. The following derivatives of OHA were synthesized: *trans*-2-octenohydroxamic acid (2-octenoHA), *trans*-3-octenohydroxamic acid (3-octenoHA), 2-octynohydroxamic acid (2-octynoHA), 3-octynohydroxamic acid (3-octynoHA), 7-octynohydroxamic acid (7-octynoHA). Overall, six saturated and five unsaturated aliphatic HAs were obtained (Fig. 1a and b). Synthesized compounds were purified by column chromatography and characterized by [1]H NMR and [13]C NMR spectroscopies. Their exact masses were determined by high-resolution mass spectrometry (HRMS). The details on the HAs synthesis procedures and characterization are provided in the Supplementary Methods.

### Activity screening
First, the urease inhibitory activity of the HA derivatives was assessed in the caecal content of Wistar rats to identify membrane-permeable

**Fig. 1 | Chemical structures of HA derivatives. a** Structures of saturated aliphatic HAs with the varying lengths of the alkyl chain that were either synthesized (heptanohydroxamic acid (HHA), nonanohydroxamic acid (NHA), decanohydroxamic acid (DHA), laurohydroxamic acid (LHA)) or purchased (acetohydroxamic acid (AHA), octanohydroxamic acid (OHA)). **b** Structures of unsaturated derivatives of OHA (*trans*-2-octenohydroxamic acid (2-octenoHA), *trans*-3-octenohydroxamic acid (3-octenoHA), 2-octynohydroxamic acid (2-octynoHA), 3-octynohydroxamic acid (3-octynoHA), 7-octynohydroxamic acid (7-octynoHA)). The structure of the lead inhibitor is shown in red.

inhibitors that were active in a complex microbial environment. Specifically, the urease inhibitory activity of the saturated HAs with alkyl chain lengths varying from 2 to 12 carbons at concentrations up to 20 mM was investigated. As most of the compounds showed limited water solubility, (2-hydroxypropyl)-β-cyclodextrin (HPβCD) or methyl-β-cyclodextrin (MβCD) were used to facilitate the dissolution of HAs in water (Supplementary Table 1). The anti-ureolytic activity was assessed by measuring the concentration of ammonia produced from urea hydrolysis by bacterial ureases within 30 min at 37 °C in the presence of increasing concentrations of inhibitors (Fig. 2a, b) or HPβCD alone (Supplementary Fig. 2). As shown in Fig. 2a, the increase of the alkyl chain length from 2 to 8 carbons led to an increase in inhibitory activity. OHA was the most potent urease inhibitor among saturated HAs with a calculated $IC_{50}$ of $0.25 \pm 0.1$ mM compared to $8.67 \pm 1.3$ and $0.50 \pm 0.2$ mM for AHA and HHA, respectively. Further increase in the chain length from 8 to 12 carbons negatively affected the potency of HAs. In fact, LHA with 12 carbons was found to be inactive in the tested range of concentrations. Interestingly, a similar relationship between the alkyl chain length and HAs activity was also observed in previous works[36,37].

As 8 carbons alkyl chain was found to be the most optimal for urease inhibition, the activity of some unsaturated derivatives of OHA was further studied (Fig. 2b). The anti-ureolytic activity of unsaturated derivatives with double or triple bonds was tested up to 1 mM and compared to that of OHA. 2-octynoHA was found to be the most potent inhibitor with the $IC_{50}$ ca. 25-fold lower compared to the $IC_{50}$ of OHA ($0.0093 \pm 0.0089$ vs. $0.25 \pm 0.1$ mM), while 3-octenoHA ($IC_{50} = 0.082 \pm 0.034$ mM), 2-octenoHA ($IC_{50} = 0.28 \pm 0.15$ mM) and 3-octynoHA ($IC_{50} = 0.11 \pm 0.03$ mM) seemed to have relatively similar activity to that of OHA. $IC_{50}$ values for all tested compounds are provided in Supplementary Table 2.

To validate the results of the caecal content-based assay, the urease inhibitory activity of 2-octynoHA, OHA and AHA was then assessed on Jack bean urease using a pH-based assay with phenol red (Supplementary Figs. 3a–c). Change in pH, which is not attributed to urea hydrolysis, was monitored in control samples in the absence of urease or in the absence of urease and urea (Supplementary Figs. 4a–f). The time ($T_{50}$) at which the absorbance value of a sample with an inhibitor is equal to half of the maximum absorbance of a corresponding control sample ($A_{1/2}$) was determined and plotted as a function of the inhibitor's concentration (Fig. 2c, Supplementary Fig. 3d). To compare inhibitors' potency, the difference in $T_{50}$ values in the absence of inhibitors (control) and with inhibitors at 0.1 mM was

calculated ($\Delta T_{50}$). As shown in Fig. 2d, 2-octynoHA was again more efficient in inhibiting ammonia production, which resulted in a longer delay in reaching the $A_{1/2}$ value ($\Delta T_{50} = 47.9 \pm 2.6$ min) compared to OHA ($\Delta T_{50} = 3.0 \pm 1.6$ min) and AHA ($\Delta T_{50} = 2.8 \pm 0.6$ min).

## In vitro characterization of 2-octynoHA

The stability of 2-octynoHA was first examined in simulated intestinal fluids (SIFs) at both pH 6.8 and pH 6.3. As shown in Fig. 2e, 2-octynoHA completely converted within 12 h of incubation at pH 6.8 and 37 °C. At pH 6.3, conversion was substantially reduced, suggesting the potential use of local acidification (through specific dosage forms) as a mean to stabilize 2-octynoHA, and hence prolong its activity. X-ray crystallographic analysis revealed that 2-octynoHA cyclized to 5-pentylisoxazol-3-ol likely due to the inherent reactivity of the hydroxamate group with the triple bond, which acts as a Michael acceptor (crystal structure and refinement data are shown in Supplementary Fig. 5 and Supplementary Table 3, respectively; formation of 5-pentylisoxazol-3-ol from 2-octynoHA in SIFs is shown in Supplementary Fig. 6a). The latter was inactive toward urease in the caecal content assay (Supplementary Fig. 6b), which might be explained by the loss of the hydroxamate group involved in the chelation of two nickel ions in the active site of urease. Interestingly, since some HAs were shown to have cytotoxic and/or mutagenic potential conditioned upon the hydroxamate group[38], the elimination of this functional group may improve the overall safety profile of 2-octynoHA, although this would require further investigation.

Cytotoxicity and mutagenicity are known concerns that have been somewhat limiting the development of HAs as therapeutic agents[38]. Since the local action of the urease inhibitor in the colon implies that intestinal epithelium is exposed to the inhibitor for a certain period of time, we assessed the viability of human colorectal adenocarcinoma cells (Caco-2) in the presence of HAs. Caco-2 cells are a well-defined intestinal cell model, which is routinely used for assessing drug permeation and toxicity[39]. Caco-2 cell viability was determined using a Cell Counting Kit-8 (CCK-8) after 24 h exposure to the increasing concentrations of 2-octynoHA, OHA and AHA and compared to the pure medium as control. As shown in Fig. 3a, 2-octynoHA did not significantly affect cell viability up to 1 mM, while at 5 and 10 mM it was reduced to $47.0 \pm 4.6$ and $5.7 \pm 3.5\%$, respectively. According to the stability study results, 2-octynoHA was expected to fully transform into 5-pentylisoxazol-3-ol within 24 h incubation period. Therefore, we assumed that cell viability after incubation with 5-pentylisoxazol-3-ol might be analogous to that of 2-octynoHA. Indeed, 5-pentylisoxazol-3-

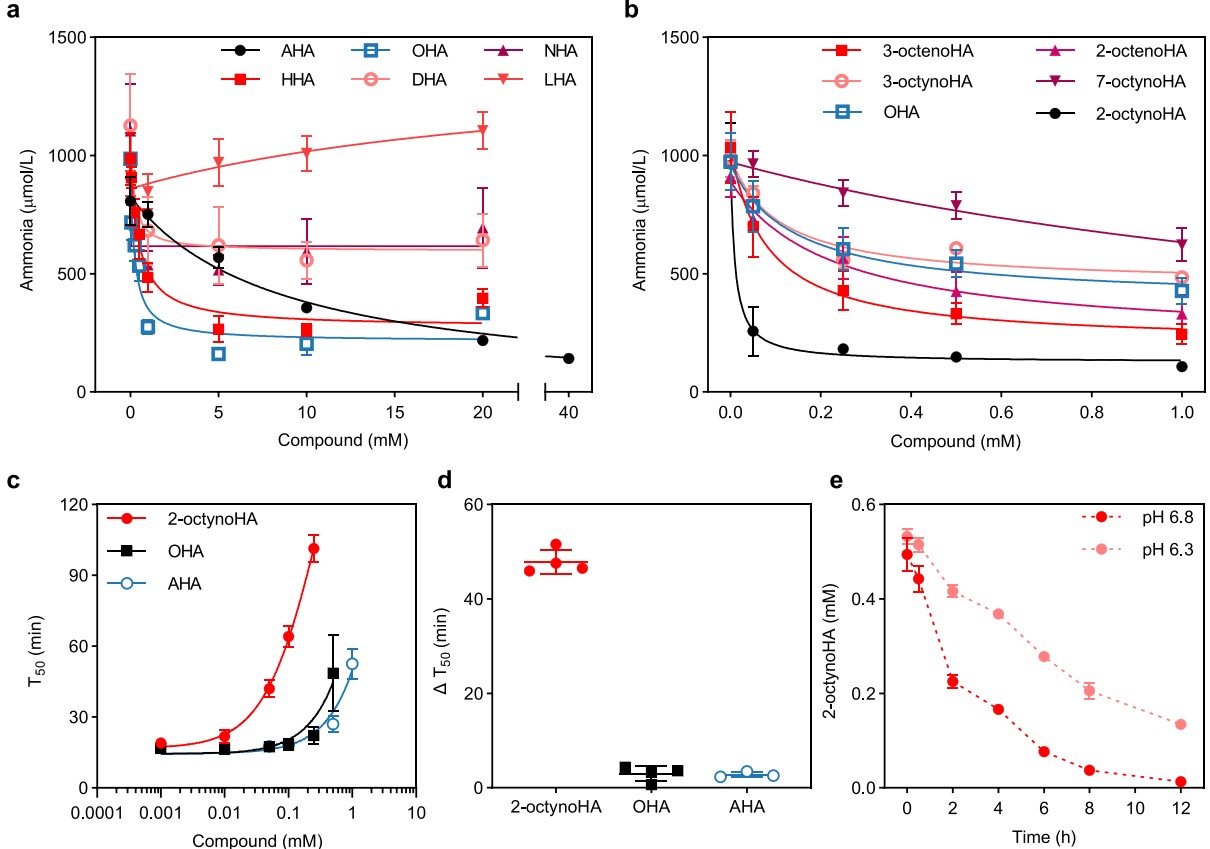

**Fig. 2 | In vitro anti-ureolytic activities of hydroxamic acid (HA) derivatives and stability of 2-octynohydroxamic acid (2-octynoHA). a, b** Concentrations of ammonia produced from urea hydrolysis in caecal content after 30 min incubation with (**a**) saturated HAs ($n = 2$ for 1 mM and 40 mM acetohydroxamic acid (AHA), $n = 3$ for 5 – 20 mM AHA; $n = 3$ for 0.05 – 20 mM heptanohydroxamic acid (HHA) and octanohydroxamic acid (OHA), $n = 6$ for the negative control (0 mM) for HHA and OHA; $n = 4$ for nonanohydroxamic acid (NHA) and decanohydroxamic acid (DHA); $n = 3$ for laurohydroxamic acid (LHA)) and (**b**) unsaturated HAs ($n = 3$ for 2-octynoHA, *trans*-2-octenohydroxamic acid (2-octenoHA), 3-octynohydroxamic acid (3-octynoHA), 7-octynohydroxamic acid (7-octynoHA); $n = 6$ for *trans*-3-

octenohydroxamic acid (3-octenoHA); $n = 11$ for OHA). **c** In vitro anti-ureolytic activities of 2-octynoHA, OHA and AHA toward Jack bean urease. $T_{50}$ is the time when the absorbance value of a sample with an inhibitor equals to half of the maximum absorbance of a corresponding control sample ($A_{1/2}$) ($n = 4$ for 2-octynoHA and OHA; $n = 3$ for AHA). **d** Activities of 2-octynoHA, OHA and AHA at 0.1 mM. $\Delta T_{50}$ is defined as the time by which $A_{1/2}$ is delayed in the presence of an inhibitor at 0.1 mM ($n = 4$ for 2-octynoHA and OHA; $n = 3$ for AHA). **e** Stability of 2-octynoHA over 12 h incubation at 37 °C in phosphate buffer at pH 6.8 and pH 6.3 ($n = 3$). Data are expressed as mean ± standard deviation (SD) from $n$ experiments. Source data are provided as a Source Data file.

ol did not affect cell viability up to 1 mM followed by its reduction to $49.1 \pm 6.1$ and $24.3 \pm 5.7\%$ at 5 and 10 mM, respectively (Supplementary Fig. 7). OHA exhibited stronger cytotoxicity significantly reducing cell viability already at 1 mM (Supplementary Fig. 8a), whereas AHA was shown to have no significant effect on cell viability up to 10 mM (Supplementary Fig. 8b).

It was previously reported that some HAs possess mutagenic properties[38,40–43], which might be related to Lossen rearrangement[38]. Indeed, metabolically activated AHA was previously shown to be mutagenic at 20 mM in V79 Chinese Hamster cells[42], whilst no mutagenic effect was reported in *Salmonella typhimurium* TM677 up to 13.3 mM[44]. Moreover, OHA exhibited a dose-dependent mutagenic activity in *Escherichia coli* wp2 hcr strain[41]. Here, the mutagenic potential of 2-octynoHA, OHA and AHA was evaluated in *S. typhimurium* TA98, TA100, TA1535, TA1537 strains and in the combination of *E. coli* wp2 [pKM101] and wp2 uvrA strains (*E. coli* Combo) after 90 min exposure by the Ames test. The effect of metabolic activation was assessed by adding S9 fraction (supernatant of rat liver homogenate). 2-octynoHA was not mutagenic up to 0.5 mM in TA98 and TA100 strains (Figs. 3b and c) as well as up to 1 mM in TA1535 and *E.coli* Combo strains (Figs. 3d and e) both in the absence and presence of S9 fraction. In TA1537 a slightly elevated number of revertant colonies was observed at 1 mM

(Fig. 3f). At higher concentrations, cytotoxic effects were observed as demonstrated by the absence of revertant colonies in all strains. Similar to 2-octynoHA, OHA was not mutagenic up to 1 mM, while at 5 and 10 mM it seemed to be cytotoxic in all strains with and without S9 fraction (Supplementary Figs. 9a–e). On the other hand, AHA, which is approved for the treatment of chronic urea-splitting urinary infection, showed a dose-dependent mutagenicity in TA98 with and without S9 fraction (Supplementary Figs. 10a–e). Moreover, a slightly elevated number of revertant colonies was observed in TA100 and TA1537 at 10 mM without S9 fraction.

In order to effectively inhibit ammonia production by urease-producing bacteria, urease inhibitors should be minimally absorbed to reach the colon at high concentrations. Therefore, we assessed the epithelial permeability of 2-octynoHA and prepared prototype colonic formulations. The apical-to-basolateral transport of 2-octynoHA was investigated using differentiated Caco-2 cell monolayers after the treatment with 0.3 mM 2-octynoHA. Due to rapid cyclization to 5-pentylisoxazol-3-ol during incubation at pH 7.4, only very low concentrations (below the lower limit of quantitation (LLOQ)) of 2-octynoHA were detected in the basolateral chamber. Therefore, the epithelial permeability of 5-pentylisoxazol-3-ol was also characterized. The cumulative fraction of 5-pentylisoxazol-3-ol transported through

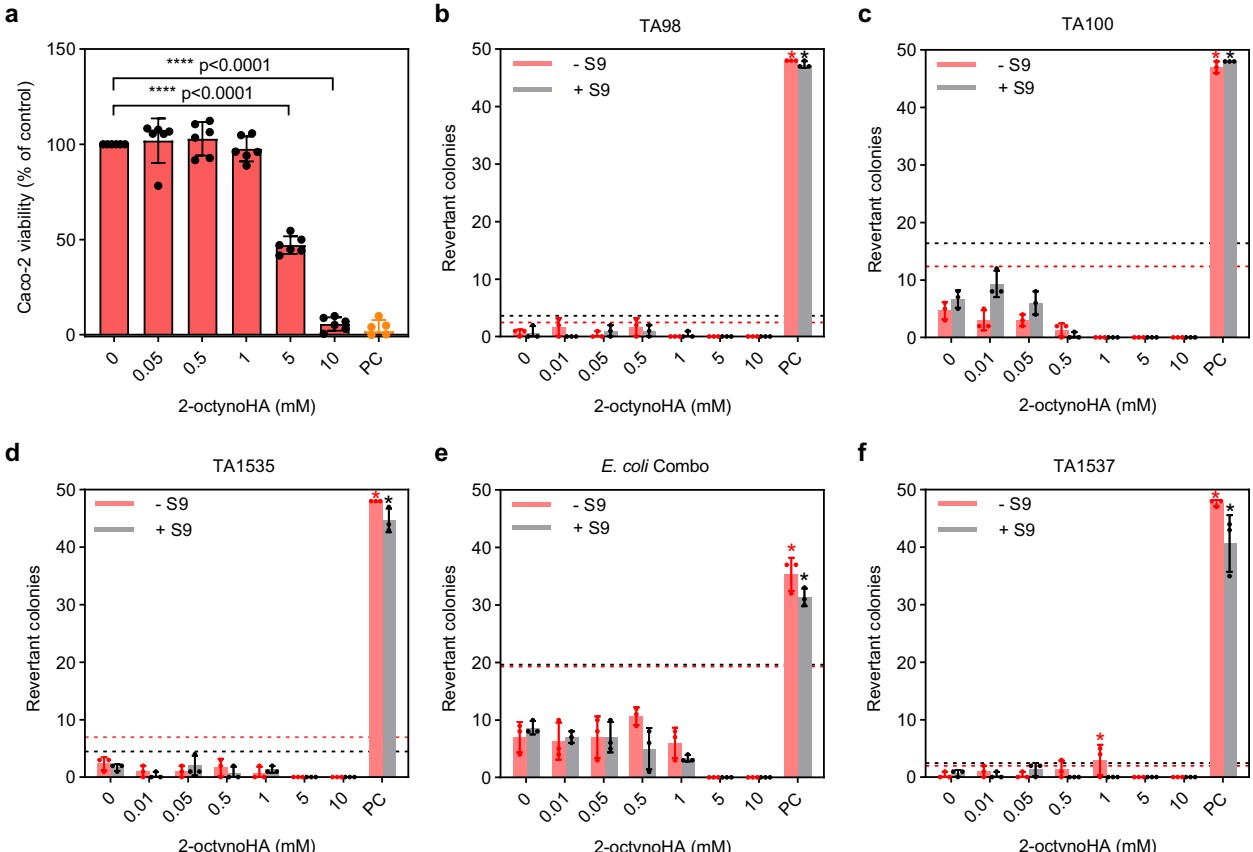

**Fig. 3 | Cytotoxicity and mutagenicity assessment of 2-octynohydroxamic acid (2-octynoHA). a** Caco-2 cell viability in the presence of 2-octynoHA compared to the medium control ($n = 6$). Positive control (PC, 10 mM hydrogen peroxide) is presented as an orange bar ($n = 6$). Statistical significance was calculated by one-way analysis of variance (ANOVA) with Tukey's multiple comparisons test with ****$p < 0.0001$ vs. medium control. All p-values are reported in Supplementary Data 1. **b–f** Mutagenicity of 2-octynoHA in (**b**) TA98 ($n = 3$), (**c**) TA100 ($n = 3$), (**d**) TA1535 ($n = 3$), (**f**) TA1537 ($n = 3$) *S. typhimurium* strains and in (**e**) a combination of *E. coli* wp2 uvrA and *E. coli* wp2 [pKM101] strains (*E. coli* Combo, $n = 3$) is expressed as a number of wells with revertant colonies per 48 wells. Mutagenicity was assessed with (+ S9) and without (- S9) liver homogenate (S9 fraction). The black and red dashed lines show a 2-fold increase over a baseline level for the experiments with and without S9 fraction, respectively, indicating the minimal number of revertant colonies per 48 wells at which the tested inhibitor's concentration is considered potentially mutagenic. Mutagenic test samples are indicated with black or red asterisk for the experiments with and without S9 fraction, respectively. Positive controls (PC) for each strain are listed in Supplementary Tables 4 and 5. Data are expressed as mean ± SD from *n* experiments. Source data are provided as a Source Data file.

the permeable supports increased linearly over 2 h (Supplementary Fig. 11). The apparent permeability coefficient ($P_{app}$) of 5-pentylisoxazol-3-ol was $5.9 \pm 1.0 \cdot 10^{-6}$ cm s$^{-1}$, which is indicative of good membrane permeability of this inactive metabolite[45].

In order to maximize the delivery of 2-octynoHA to the colon, enteric capsules that would withstand low pH of the upper GI tract and dissolve in the lower GI tract in the dog and rat models were prepared. A series of coatings composed of various methacrylic acid copolymers and 5% (w/w) triethyl citrate (TEC) as plasticizer was investigated (see the Methods section).

As the pH values of the canine small intestine were reported to be neutral or slightly basic[46], Eudragit S100 (dissolution pH of around 7) was selected for the capsule coatings. As shown in Supplementary Table 6, increasing concentrations of Eudragit S100, as well as the number of coating layers led to the increased resistance of the capsules (size 0) in both simulated gastric fluid (SGF) and SIF. Thus, the most optimal disintegration profile was achieved with capsules coated with two layers of 15% (w/w) Eudragit S100. In the in vitro disintegration test, these capsules remained resistant in SGF for at least 3 h, but generally started to disintegrate at ca. 2.5 h, which corresponds to the transit time of the small intestine (ca. 2–3 h) in dogs[46].

In rats, since the intestinal environment is slightly acidic[47], enteric capsules (size 0) that disintegrate at pH 5.5 within the range of the small intestine transit time reported for rats (ca. 100 min[48]) were prepared (Supplementary Table 7). A coating composed of two layers of 15% (w/w) Eudragit L100 generated capsules that were resistant for at least 3 h in SGF but that disintegrated within 1 h at pH 5.5.

**Pharmacokinetics of 2-octynoHA in dogs**

The pharmacokinetics (PK) of 2-octynoHA was assessed in beagle dogs following the I.V. administration of a solution of the compound with HPβCD (PK 1) and its peroral (P.O.) administration in an uncoated gelatin capsule (PK 2) or the enteric capsule (PK 3) (Supplementary Table 8). After a single I.V. dose of 100 mg, 2-octynoHA displayed quantifiable levels (above LLOQ) up to 1.5 h post-dosing (Fig. 4a), with a short half-life time ($t_{1/2}$) of $0.17 \pm 0.01$ h (Fig. 4c). The maximum plasma concentration ($C_{max}$) at 0.083 h post-injection (first measured time point) and the area under the concentration vs. time curve ($AUC_{0-t}$) were 6246 ng/mL and $AUC_{0-1.5} = 1773$ ng/mL·h, respectively (Fig. 4c). When 300 mg 2-octynoHA were administered orally with the uncoated capsule, plasma levels of the compound were measurable up to 6 h post-dosing (Fig. 4b). The $C_{max}$ and $AUC_{0-6}$ were 41 ng/mL and 167 ng/mL·h, respectively (Fig. 4c). The oral bioavailability of 2-octynoHA was

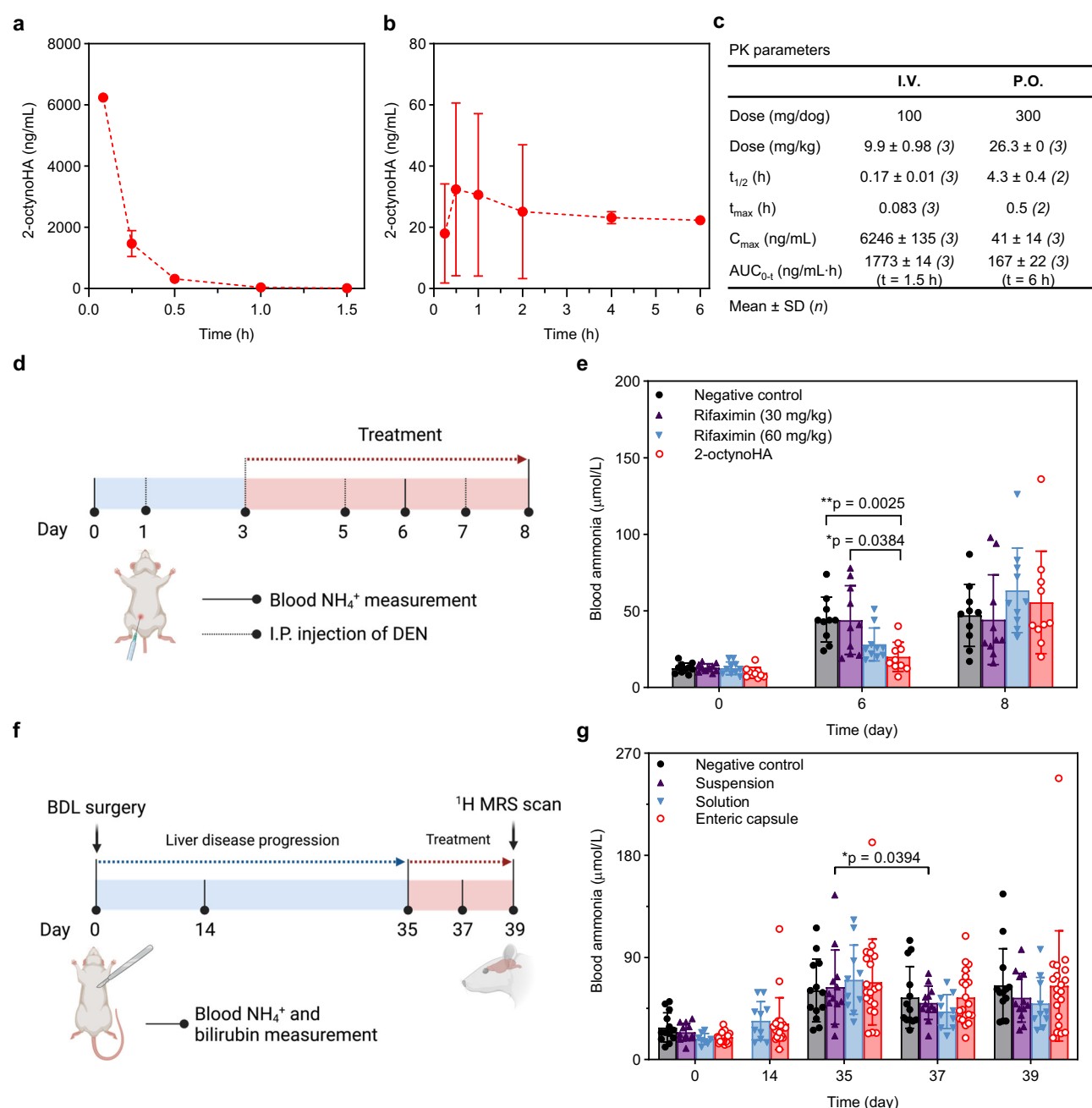

**Fig. 4 | Pharmacokinetics (PK) and in vivo efficacy of 2-octynohydroxamic acid (2-octynoHA). a, b** 2-octynoHA plasma concentration profiles in dogs following (**a**) intravenous (I.V.) injection (*n* = 3) and (**b**) oral administration (P.O.) of an uncoated capsule (*n* = 3). **c** PK parameters of 2-octynoHA (*n* = 3 for all parameters except for $t_{1/2}$ and $t_{max}$ in the P.O. group, *n* = 2 for $t_{1/2}$ and $t_{max}$ in the P.O. group). **d** In vivo study design in a rat model of N-nitrosodiethylamine (DEN)-induced liver injury. Rats received intraperitoneal (I.P.) injections of DEN on days 1, 3, 5 and 7 and treatment from day 3 to 8 twice a day. Blood ammonia was monitored on days 0, 6 and 8. **e** Blood ammonia levels in DEN rats treated with (2-hydroxypropyl)-β-cyclodextrin (HPβCD, negative control, *n* = 10), 15 mg/kg 2-octynoHA (solution with HPβCD, *n* = 10), 30 mg/kg (*n* = 10) or 60 mg/kg (*n* = 10) rifaximin. Statistical significance between groups was calculated by two-way repeated measures ANOVA with Tukey's multiple comparisons test with \**p* < 0.05, \*\**p* < 0.01. All p-values are

reported in Supplementary Data 2. **f** In vivo study design in a bile duct ligated (BDL) rat model. Rats received treatment from day 35 to 39 post-surgery. Blood ammonia and bilirubin were monitored before BDL surgery (day 0), on days 14, 35, 37 and 39 post-surgery. Brain metabolites were measured using in vivo ¹H MRS on day 39. **g** Blood ammonia levels in BDL rats in the negative control (drug-free solution, *n* = 13), 30 mg/kg 2-octynoHA suspension (*n* = 12), and 30 mg/kg 2-octynoHA solution (with HPβCD, *n* = 8 on day 0, *n* = 10 on days 14–39) groups treated twice daily, and in the 10 mg 2-octynoHA enteric capsule group treated once daily. Statistical significance between treatment days (35–39) of each group was calculated by one-way repeated measures ANOVA with Tukey's multiple comparisons test with \**p* < 0.05. All p-values are reported in Supplementary Data 3. Data are expressed as mean ± SD from *n* animals. Source data are provided as a Source Data file.

found to be ca. 3.5%. This might reflect the rapid conversion of 2-octynoHA into 5-pentylisoxazol-3-ol and/or limited dissolution of 2-octynoHA since the capsules did not contain any solubilizing agent.

When 2-octynoHA was with the enteric capsule, the plasma levels were below LLOQ in almost all animals indicating lower systemic

exposure compared to the control capsule formulation. However, by monitoring several metabolites of 2-octynoHA (Supplementary Table 9), we found that the enteric capsule delayed and reduced the systemic exposure to some of these metabolites (Supplementary Fig. 12). These data might indicate that delivery of 2-octynoHA in the

enteric capsule led to the delayed release of the compound and/or reduced absorption, yet this requires further investigation.

### Efficacy of 2-octynoHA in a rodent model of acute liver injury

Considering the ammonia-lowering mechanism of action of 2-octynoHA, an animal model manifesting hyperammonemia regardless of the origin of liver disease was first chosen in order to assess the inhibitor's in vivo efficacy. Thus, the ability of 2-octynoHA to reduce systemic ammonemia was assessed in a rat model of acute liver injury induced by recurrent intraperitoneal injections (I.P.) of N-nitrosodiethylamine (DEN)[49,50], and was compared to that of rifaximin. Animals received DEN injections every two days from day 1 to 7, and staring from day 3 of the study, they were gavaged twice a day with 2-octynoHA (15 mg/kg in a HPβCD solution), rifaximin (30 mg/kg or 60 mg/kg) or the negative control (HPβCD solution) (Fig. 4d). Regular injections of DEN led to the increase in blood ammonia levels (Fig. 4e) and decreased body weight in all groups possibly due to the hepatotoxicity of DEN (Supplementary Fig. 13). Interestingly, a significant decrease in ammonemia was observed on day 6 in the group receiving 2-octynoHA ($20.0 \pm 9.6$ μmol/L) compared to the negative control group receiving HPβCD ($44.3 \pm 14.7$ μmol/L). On the opposite, rats receiving 30 mg/kg or 60 mg/kg rifaximin had average blood ammonia levels on that day that were not significantly different from that of the control group ($44.0 \pm 22.4$ and $28.1 \pm 10.8$ μmol/L, respectively). On day 8, all groups displayed increased and comparable blood ammonia levels, which might be explained by the progression of liver damage induced by DEN. Indeed, a higher number of DEN injections on day 8 likely led to a more severe deterioration of liver function and thereby ammonia metabolism compared to day 6. Although ammonia production via urea hydrolysis might have been inhibited on both days, it has not been sufficient to impact the ammonemia on day 8.

### Efficacy of 2-octynoHA in a rodent model of chronic liver disease

The in vivo efficacy of 2-octynoHA was also investigated in a rat model of type C HE induced by obstruction of the common bile duct[51] (performed on day 0) (Fig. 4f). On day 35 after the bile duct ligation (BDL), both blood ammonia (Fig. 4g) and plasma bilirubin levels (Supplementary Fig. 14) markedly increased in BDL rats compared to those before surgery, confirming liver damage. Starting from day 35, animals received a 5-day treatment of 2-octynoHA either as aqueous suspension (30 mg/kg, twice daily), aqueous HPβCD solution (30 mg/kg, twice daily) or enteric capsule (10 mg or ca. 30 mg/kg, daily), or a control drug-free aqueous solution (Fig. 4f). In all treated groups, there was a trend towards reduced blood ammonia levels after three days of treatment (day 37) with statistical significance achieved in the group treated with the 2-octynoHA solution ($42.3 \pm 14.95$ μmol/L vs. $70.4 \pm 30.6$ μmol/L, corresponding to a decrease in ammonemia of ca. 40%; multiplicity adjusted $p = 0.0394$) (Fig. 4g). On day 39, the effect was less pronounced, especially in the groups receiving the suspension or enteric capsules. This might be explained by the progression of the disease, the insufficient dissolution of the compound in the large intestine, and in the case of the capsule, by the lower administered dose. As expected, bilirubin levels did not seem to be affected by 2-octynoHA treatment (Supplementary Fig. 14). Bilirubin was comparably high in all 2-octynoHA-treated groups during treatment days indicating liver insufficiency, which could not be addressed by a urease inhibitor due to its mechanism of action. Animals receiving 2-octynoHA formulated as a solution or as an enteric capsule showed a statistically significant reduction of the body weight on days 37 and/or 39 post-surgery corresponding to the third and fifth days of the treatment, respectively, which was within 10% difference compared to the body weight at the beginning of the treatment (day 35) (Supplementary Fig. 15). The origin of this effect is unclear and may be related to the stress induced by the animal handling (i.e. oral gavage, blood sampling) and decreased food consumption. Plasma glucose (Glc)

levels seem to decrease throughout the study, which might indicate the declining health status of animals subjected to BDL (Supplementary Fig. 16).

To further investigate the effects of 2-octynoHA in BDL rats, we conducted in vivo $^1$H magnetic resonance spectroscopy ($^1$H MRS) measurements of 14 brain metabolites following the last treatment administration on day 39. The metabolites were assessed in the cerebellum as this region was previously shown to exhibit more distinct changes (i.e. in Gln) compared to the hippocampus in HE in BDL rats[52,53]. A significant 28% decrease in brain Gln was observed in the group treated with 2-octynoHA-based solution compared to the negative control group ($5.1 \pm 1.1$ vs. $7.1 \pm 1.2$ mmol/kg$_{ww}$, multiplicity adjusted $p = 0.0019$) (Fig. 5a, the representative spectra are shown in Supplementary Fig. 17). Interestingly, a 18% decrease trend in brain Gln was also observed in the enteric capsule group ($5.8 \pm 0.99$ vs. $7.1 \pm 1.2$ mmol/kg$_{ww}$, multiplicity adjusted $p = 0.0643$), while a 15% decrease trend in Gln was found in the suspension group compared to the negative control group ($6.0 \pm 0.9$ vs. $7.1 \pm 1.2$ mmol/kg$_{ww}$, multiplicity adjusted $p = 0.1387$). As Gln synthesis is the main pathway of ammonia detoxification in astrocytes, brain Gln levels are strongly correlated with circulating ammonia[9,52]. Therefore, the observed reduction in Gln might be related to inhibited urease activity and thereby ammonia production in the presence of 2-octynoHA.

Previous studies on BDL rats showed that osmotic stress created by Gln accumulation might be in part offset by the decrease in other organic osmolytes[52–55]. Therefore, in the opposite case, the increase in main osmolytes as a response to decreased Gln was anticipated. However, herein in light of decreased Gln in the 2-octynoHA solution group, organic brain osmolytes such as taurine (Tau), creatine (Cr), phosphocreatine (PCr) and total creatine (tCr) significantly decreased. Myo-inositol (Ins) and total choline (tCh) were comparable to that of the negative control group (Figs. 5b–g). Specifically, Tau and Cr decreased by 21%, while PCr and tCr decreased by 15% and 18%, respectively (Figs. 5c–e). Interestingly, similar results were obtained for the group receiving 2-octynoHA capsules. Thus, a significant decrease in Cr (−17%) and tCr (−14%) levels was found in the enteric capsule group compared to the control group (Figs. 5d and f). Overall, alterations in brain osmolytes were more evident in the group treated with a 2-octynoHA solution. Although the underlying reason behind decreased osmolytes in the solution and enteric capsule group is unclear, possible explanations may involve delayed osmotic compensation for the Gln decline[56–58], insufficient treatment duration[52,59] or additional osmotic stress driving the outflow of these osmolytes.

Regarding the neurotransmitters, glutamate (Glu) significantly decreased by 22% and 20% in the solution and capsule groups compared to the negative control, respectively, whereas γ-aminobutyric acid (GABA) decreased by 32% only in the solution group (Figs. 5h and i). The glutamatergic and GABAergic neurotransmission systems are known to be affected in HE[60]. Previous works in BDL rats reported decreased cerebellar Glu[52,53], while GABA levels in the cerebellum were either stable[52] or decreased[53]. Herein, the reasons behind the observed decrease in Glu and GABA compared to the negative control and the role of 2-octynoHA in this process remain unclear warranting further investigation.

Brain Glc levels were lower in the solution (−53%) and capsule (−45%) groups compared to the negative control (Supplementary Fig. 18a), which might reflect lower plasma Glc levels in these groups (Supplementary Fig. 16). No significant differences were observed in ascorbate (Asc), lactate (Lac), N-acetylaspartylglutamate (NAAG) and phosphoethanolamine (PE) levels in 2-octynoHA solution and capsule groups compared to the negative control (Supplementary Figs. 18b–e). Asc acts as an antioxidant which was previously shown to gradually decrease over the course of HE development in BDL rats[52,54,61]. Lac is an energy metabolite for which a sudden increase at the late stages of the disease was reported by earlier studies[52,61]. As

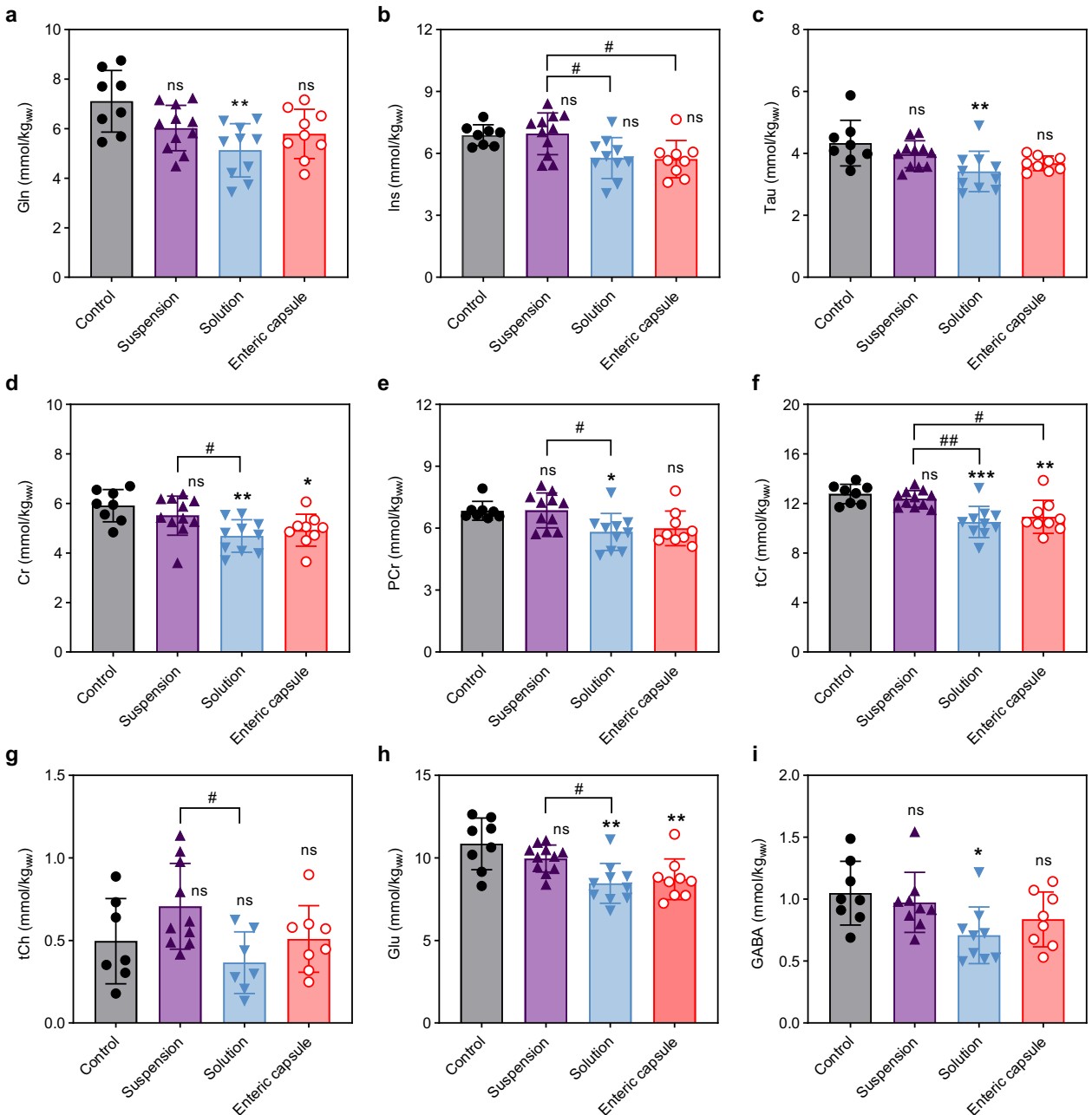

**Fig. 5 | Brain metabolites levels in bile duct ligated (BDL) rats measured by ¹H MRS. a–i** Concentrations of (**a**) glutamine (Gln), (**b**) myo-inositol (Ins), (**c**) taurine (Tau), (**d**) creatine (Cr), (**e**) phosphocreatine (PCr), (**f**) total creatine (tCr), (**g**) total choline (tCh), (**h**) glutamate (Glu) and (**i**) γ-aminobutyric acid (GABA) after a 5-day treatment with the drug-free aqueous solution (control), 2-octynoHA (30 mg/kg suspension, twice daily), 2-octynoHA (30 mg/kg solution of HPβCD, twice daily), 2-octynoHA (10 mg in an enteric capsule, once daily). (**a–f, h**) $n = 8$, $n = 11$, $n = 10$, $n = 9$ for the control, suspension, solution and capsule groups, respectively. (**g**) $n = 7$, $n = 10$, $n = 7$, $n = 8$ for the control, suspension, solution and capsule groups,

respectively. (**i**) $n = 8$, $n = 9$, $n = 9$, $n = 8$ for the control, suspension, solution and capsule groups, respectively. Due to the length and complexity of ¹H MRS measurements, not all animals in each group underwent the measurements. Statistical significance between the groups was calculated by one-way ANOVA with Tukey's multiple comparisons test with *$p < 0.05$, **$p < 0.01$, ***$p < 0.001$; ns – not significant vs. control; statistical significance for other comparisons is indicated with #$p < 0.05$, ##$p < 0.01$. All $p$-values are reported in Supplementary Data 4–12. Data are expressed as mean ± SD from $n$ animals. Source data are provided as a Source Data file.

for NAAG and PE, both molecules were shown to be unresponsive to HE[52].

Interestingly, almost all metabolites seemed to be unaffected by the treatment with 2-octynoHA suspension (Fig. 5, Supplementary Figs. 18a–e) except for a neuronal marker, N-acetylaspartate (NAA), which was significantly decreased in all three 2-octynoHA-treated groups (Supplementary Fig. 18f). The mechanism behind decreased

NAA is uncertain, however, its alterations might be associated with osmotic regulation as well as compromised neuronal function[62].

In summary, we identified a hydroxamate-based urease inhibitor, 2-octynoHA, which exhibited a potency exceeding that of HAs previously tested in HE patients (e.g. AHA and OHA). Although the stability tests showed that the 2-octynoHA activity under conditions simulating the GI tract might be limited by its cyclization to 5-pentylisoxazol-3-ol,

the slight acidification of the medium could greatly decrease the conversion rate. These findings suggest that the 2-octynoHA formulation may be improved by including acidifying excipients (e.g. citric acid). On the other side, the loss of the hydroxamate group might be advantageous as HAs have been associated with potential mutagenicity[38]. While the long-term safety of 2-octynoHA remains to be assessed, the compound was not found cytotoxic nor mutagenic within a micromolar concentration range. Moreover, if delivered to the colon, low absorption and thereby low risk of systemic adverse effects are anticipated as suggested by the dog bioavailability data.

In rats, the aqueous solution of 2-octynoHA containing HPβCD significantly reduced ammonemia in both the DEN-induced liver disease model and the BDL model of type C HE. However, in both models, reduced ammonemia was observed only midway through the treatment which was then followed by the increase in ammonia levels at the end of the treatment. It is worth mentioning that such a trend was also found in the DEN model groups treated with rifaximin suggesting that it might be attributed to the late-stage liver disease and thereby to dramatically reduced liver capacity to metabolize ammonia. Therefore, further research should focus on assessing the efficacy of 2-octynoHA treatment when introduced earlier. Nevertheless, despite a slight elevation in ammonia levels, a significant decrease in cerebellar Gln in BDL rats treated with 2-octynoHA solution compared to the control was observed at the end of the treatment which additionally supports the effectiveness of 2-octynoHA in inhibiting ammoniagenesis. Interestingly, along with the decreased Gln level, other brain metabolites (i.e. Tau, Cr, PCr, tCr, Glu, GABA, Glc, NAA) also decreased compared to the negative control group. The mechanism behind these changes as well as practical implications are unclear, and therefore, should be addressed in further studies involving longitudinal ¹H MRS experiments and behavioral tests. Furthermore, the effects of 2-octynoHA suspension or enteric capsules on hyperammonemia and neurometabolic profile of BDL rats were less pronounced than that of a solution highlighting the importance of a dissolution agent in the formulation.

Taken together, this work highlights 2-octynoHA as a promising therapeutic option for HE. Future research should aim at maximizing 2-octynoHA activity in the colon through formulation optimization, characterizing its safety profile and assessing its efficacy in reducing neuropsychiatric symptoms of HE.

## Methods

Research in this manuscript complies with all relevant ethical regulations. Animal procedures in the studies with dogs were approved by the Comité Institutionnel de Protection des Animaux (CIPA) of the Institut National de la Recherche Scientifique (INRS) (Ethical Animal Protocol #2002-04). The INRS test facility is accredited by Association for Assessment and Accreditation of Laboratory Animal Care (AAALAC) and Canadian Council on Animal Care (CCAC). Animal experiments in a rat model of DEN-induced liver injury were approved by the Institutional Animal Care and Use Committee (IUCAC) of Wuhan Servicebio Technology Co., Ltd. (protocol numbers: 2021071, 2021136, 2021137, 2022003). Animal experiments in BDL rats were conducted according to the Federal and local ethical guidelines, the protocols were approved by the local Committee on Animal Experimentation for the Canton de Vaud, Switzerland (VD3022.1). Animal experiments in this manuscript are reported according to the ARRIVE guidelines.

### Materials

All materials are listed in the Supplementary Methods.

### Synthesis and characterization of HAs

The synthesis procedures for HAs and production of 5-pentylisoxazol-3-ol are described in detail in the Supplementary Methods. The compounds were characterized by ¹H NMR and ¹³C NMR (400 MHz, AV400 spectrometer, Bruker). Exact masses of HHA, NHA, DHA, LHA, 2-octenoHA, 3-octenoHA, 2-octynoHA, 3-octynoHA and 7-octynoHA were determined by high-resolution mass spectrometry (HRMS) (electrospray ionization (ESI)-quadrupole-time-of-flight (QTOF) tandem mass spectrometry; maXis, Bruker Daltonics). Exact mass of 5-pentylisoxazol-3-ol was determined by gas chromatography-mass spectrometry (Trace 1310 GC with the Q Exactive GC, Thermo Scientific) using TraceGold TG-5Silms GC column (30 m length, 0.25 mm inner diameter, 0.25 μm film thickness, Thermo Scientific).

### X-ray crystallography

Crystals of 5-pentylisoxazol-3-ol were grown by slow evaporation of ethyl acetate from ethyl acetate/cyclohexane (1:1, v/v) mixture. A single crystalline sample was measured using a Rigaku Oxford Diffraction XtaLAB Synergy-S Dualflex kappa diffractometer equipped with a Dectris Pilatus 300 HPAD detector and using microfocus sealed tube Mo·Kα radiation with mirror optics. Measurements were performed at 100 K using an Oxford Cryosystems Cryostream 800 sample cryostat. Data were integrated using CrysAlisPro 1.171.41.119a (Rigaku Oxford Diffraction, 2022) and corrected for absorption effects using a combination of empirical (ABSPACK) and numerical corrections. The structure was solved with the SHELXT[63] structure solution program using intrinsic phasing and refined with the SHELXL[64] refinement package using least squares minimization in Olex2[65]. Crystallographic data have been deposited at the Cambridge Crystallographic Data Centre under deposition number: 2257012 (https://doi.org/10.5517/ccdc.csd.cc2frlvn).

### In vitro caecal content-based screening assay

Urease inhibitory activity of HAs was evaluated in caecal content of Wistar rats stored at −80 °C right after collection. The caecal samples were a donation from the ETH Phenomics Center. Before the experiment, the samples were thawed at 37 °C, pooled together and diluted with 200 mM $KH_2PO_4$ (pH 6.8) to ca. 5% w/v. The mixture was centrifuged for 1 min at 100 x $g$ at RT to precipitate large particles. The supernatant with dispersed bacteria was aliquoted and stored at −80 °C until use. Due to heterogeneity of caecal content, urease activity across different pools was standardized by diluting supernatant thawed at 37 °C with 200 mM $KH_2PO_4$ (pH 6.8) to obtain suspension of bacteria producing ammonia to the final concentration of ca. 1000 μM after 30 min incubation with urea.

Due to the limited water solubility of some HAs, HPβCD and MβCD in molar ratio 1:4 as well as dimethyl sulfoxide (DMSO) were used to prepare respective solutions of HAs for the assay (Supplementary Table 1).

To measure the anti-ureolytic activity of HAs and HPβCD alone, 750 μL aliquots of caecal bacteria suspension with adjusted urease activity were centrifuged at 12,000 x $g$ for 2 min. Aliquots of supernatant were collected for quantification of background ammonia concentration. The pellets were gently resuspended and incubated with an increasing range of HAs concentrations and urea (30–60 mM) for 30 min at 37 °C with shaking. After incubation, samples were centrifuged at 12,000 x $g$ for 2 min, and aliquots of supernatant were collected for ammonia quantification.

Ammonia concentrations before and after incubation were determined using an enzymatic assay (AM1015, Randox Laboratories) as described earlier[66]. Two hundred microliters of the reagent solution were mixed with 20 μL of a test sample in a 96-well plate (flat bottom, PS; Greiner Bio-One). After 5 min of incubation at RT, the sample's absorbance was measured at 340 nm using a plate reader (Infinite M200, Tecan). Two microliters of the enzyme (glutamate dehydrogenase) were added to a well, mixed by pipetting and left to incubate for 5 min. The following decrease in sample absorbance was recorded at 340 nm. Ammonia concentrations in all samples were

calculated according to the manufacturer's instructions. Overall ammonia produced due to urea hydrolysis was quantified as the difference of volume-adjusted ammonia concentrations before and after incubation with urea and HAs. Ammonia concentrations were plotted against inhibitors' concentrations, $IC_{50}$ values were determined by nonlinear regression curve fitting using GraphPad Prism version 9.4.1 for Windows, ([inhibitor] vs. response, three parameters mode, GraphPad Software).

### pH-based cell-free urease activity assay

Urease inhibitory activities of 2-octynoHA, OHA and AHA were tested in a previously described pH-based urease activity assay with minor modifications[67]. The assay contained Jack bean urease, urea as a substrate of urease, phenol red as a pH indicator, and the inhibitor of interest in 15 mM $KH_2PO_4$ (pH 6.8). A stock solution of urease 2 mg/mL (51.8 U/mL) was prepared in 15 mM $KH_2PO_4$ (pH 6.8), while stock solutions of 0.2 mg/mL phenol red, 1 M urea and 10 mM HAs were prepared in ultra-pure water. In the 96-well plate, a mixture of 0.01 mg/mL phenol red and 100 mM urea in 15 mM $KH_2PO_4$ (pH 6.8) was supplemented with the inhibitor. 2-octynoHA and OHA were tested in the range of concentrations from 0.001 to 1 mM. AHA was tested at concentrations varying from 0.05 to 20 mM. Once an aliquot of urease was added to a well at a final concentration of 0.1 mg/mL (2.6 U/mL), the absorbance at 560 nm ($OD_{560}$) was measured every 60 s for 2 h at 37 °C using a plate reader (Infinite M200, Tecan). An $OD_{560}$ vs. time graph was generated per each tested inhibitor concentration and negative control samples without an inhibitor. Half of the maximum absorbance ($A_{1/2}$) of the control samples was calculated for each biological replicate using Eq. (1):

$$A_{1/2} = \frac{(A\max - A\min)}{2} + A\min \tag{1}$$

where $A_{max}$ is a maximum response of the control sample, $A_{min}$ is a minimum response of the control sample.

The respective time ($T_{50}$) required to achieve $A_{1/2}$ value of the control sample in the presence of inhibitors was identified by applying linear regression analysis to two data points in the proximity of the target $A_{1/2}$ value. To compare the activities of HAs, a $T_{50}$ (min) vs. inhibitor concentration (mM) graph was generated, the nonlinear regression curve fitting using GraphPad Prism version 9.4.1 for Windows, ([inhibitor] vs. response, three parameters mode, GraphPad Software) was applied.

### Stability assay

Solutions of 10 mM 2-octynoHA were prepared in 50 mM $KH_2PO_4$ buffer solution either at pH 6.8 or pH 6.3. Samples were incubated at 37 °C with shaking for 24 h. Aliquots of 50 μL were collected at 0.5, 2, 4, 6, 8, and 12 h and transferred into vials containing the internal standard OHA and $H_3PO_4$ in ultra-pure water to final concentrations of 0.5 mM and 0.015% (v/v), respectively. Samples were vortexed and stored at −20 °C until further analysis. 2-octynoHA and 5-pentylisoxazol-3-ol were detected and quantified by LC-UV/Vis.

### Quantification of 2-octynoHA by LC-UV/Vis

Samples collected in the stability assay were analyzed using the Ulti-Mate 3000 UHPLC system (Thermo Fisher Scientific) equipped with the UltiMate 3000 RS Diode Array detector (Thermo Fisher Scientific). Separation was performed using a Kinetex Polar C18 column (2.1 × 100 mm, 2.6 μm, 100 Å) (Phenomenex) equipped with a C18 polar pre-column (Phenomenex) at 25 °C. Samples were injected at a volume of 2 μL, and a flow rate was set to 0.4 mL/min. The mobile phase was composed of eluent A ($H_2O$ + 0.1% (v/v) formic acid (FA)) and eluent B (ACN + 0.1% (v/v) FA). The following gradient program was used: equilibration for 2 min at 5% B, to 100% B in 8 min, 3 min at 100% B, and

to 5% B in 2 min. UV absorbance was monitored at 210 nm. Peak integration was performed using the Chromeleon 7.2.9 software (Thermo Fischer Scientific).

To quantify 2-octynoHA, a stock solution of 10 mM 2-octynoHA in ultra-pure water was diluted to obtain eight calibrants in the concentration range of 0.0032 to 2 mM. Calibrants were prepared in triplicates from independent stock solutions and were analyzed as described above. The calibration curve was fitted using simple linear regression. All calibration samples were compared to their theoretical value. Quantitative accuracy for the lowest calibration sample (0.0032 mM) was 25%, while for the rest calibration samples it was 10% of the target. Three quality control (QC) samples (0.008, 0.08 and 1 mM) were prepared in triplicates from independent QC stock solutions of 10 mM 2-octynoHA in $H_2O$. The method was validated by calculating the accuracy (bias) and precision (relative standard deviation (RSD)) of the calculated QC concentrations which were within ± 15% (Supplementary Table 11). The method was applied to determine 2-octynoHA concentrations only within the range of the lowest and the highest QCs.

To quantify 5-pentylisoxazol-3-ol, a stock solution was prepared by incubating 10 mM 2-octynoHA in 200 mM $KH_2PO_4$ (pH 6.8) at 37 °C until complete degradation of 2-octynoHA. Eight calibrants in the concentration range of 0.04 to 2 mM were prepared from the 10 mM 5-pentylisoxazol-3-ol stock solution using ultra-pure water and were analyzed as described above. Simple linear regression analysis was used to fit the calibration curve. Calibration samples were compared to their theoretical value. Quantitative accuracy for the lowest calibration sample (0.04 mM) was 30%, while for the rest calibration samples it was 10% of the target. Three QCs (0.04, 0.08, 1 mM) were prepared in triplicates from independent stock solutions. The accuracy (bias) and precision (relative standard deviation, RSD) of the calculated QC concentrations were accurate within ± 20% (Supplementary Table 11). The method was applied to determine 5-pentylisoxazol-3-ol concentrations only within the range of the lowest and the highest QCs.

### Mutagenicity assay

Mutagenicity of AHA, OHA and 2-octynoHA was assessed using the Ames MPF™ Penta 1 microplate format mutagenicity assay kit (Xenometrix AG). Four *S.typhimurium* strains (TA98, TA100, TA1535, TA1537) and the mixture of two *E.coli* strains (wp2 [pKM101] and wp2 uvrA) were used.

The assay was performed according to the manufacturer's instructions. Briefly, serial dilutions of AHA, OHA and 2-octynoHA were prepared in DMSO, 10 μL aliquots of each concentration of the test compounds as well as a negative control (DMSO) and solutions of positive control (PC) substances provided in the kit (Supplementary Tables 4 and 5) were placed into a 24-well plate (TPP Tissue Culture Test Plate, TPP Techno Plastic Products AG). Overnight bacterial cultures were diluted (1:10) with medium containing histidine or tryptophan for *S. typhimurium* and *E. coli* strains, respectively. A volume of 240 μL of cell suspensions was added to the wells with corresponding treatments. Bacteria were incubated with the test compounds for 90 min at 37 °C with shaking at 250 rpm (TH 30, Edmund Bühler). After incubation, cultures were diluted with a pH indicator medium depleted in histidine or tryptophan and were aliquoted into 384-well plates (flat bottom, PS; BRAND). The 384-well plates were placed into the sealable plastic incubation bag provided in the kit and left for 2-day selection incubation at 37 °C. Medium with mutated cells that reverted to amino acid prototrophy turned from purple to yellow as bacteria metabolism reduced the pH of the media. The number of yellow wells containing revertant colonies was counted for each concentration of the test compounds and compared to the number of spontaneous revertant colonies of the negative control. The compound was considered mutagenic if the number of revertant colonies was at least twice higher than that of the negative control. Each concentration of

the test compounds was tested in triplicate in the presence or absence of rat liver homogenate (S9 fraction, S9 Cofactor Kit, Xenometrix AG).

## Cell culture

Human colorectal adenocarcinoma cells (Caco-2, American Type Culture Collection (ATCC), HTB-37) were cultured at 37 °C in a humidified atmosphere with 5% $CO_2$ in a complete medium containing Dulbecco's Modified Eagle Medium (DMEM, high glucose, GlutaMax, pyruvate; Thermo Fisher Scientific) supplemented with 10% (v/v) fetal bovine serum (FBS, Thermo Fisher Scientific), 1% (v/v) penicillin-streptomycin (Thermo Fisher Scientific) and 15 mM HEPES (Thermo Fisher Scientific). Caco-2 cells were used in the experiments from passages 50 to 60, and were regularly tested for mycoplasma using the MycoAlert PLUS Mycoplasma Detection Kit (Lonza).

## Cytotoxicity assay

The cytotoxicity of AHA, OHA, 2-octynoHA and 5-pentylisoxazol-3-ol was assessed using CCK-8 (Sigma-Aldrich) following the manufacturer's protocol. Caco-2 cells were seeded in a 96-well plate (TPP Tissue Culture Test Plate, TPP Techno Plastic Products AG) in a complete medium (DMEM (high glucose, GlutaMax, pyruvate), 10% (v/v) FBS, 1% (v/v) penicillin-streptomycin, 15 mM HEPES) at a density of $5 \times 10^3$ cell/well and left to attach for 24 h. The cell medium was gently removed and replaced by a complete medium containing various concentrations of tested compounds (0.05, 0.5, 1, 5, 10 mM). As positive and negative controls, a complete medium with or without 10 mM hydrogen peroxide were utilized, respectively. After 24 h incubation with compounds, cells were washed with PBS (pH 7.4, Thermo Fisher Scientific) and treated with DMEM without phenol red (Thermo Fisher Scientific) containing a CCK-8 reagent. The absorbance values were measured at 450 nm after 2–4 h incubation with a CCK-8 reagent using a plate reader (Infinite M200, Tecan). Caco-2 cell viability was calculated as a percentage of the medium-only control.

## Permeability assay

The permeability of 2-octynoHA was assessed using differentiated Caco-2 cell monolayers cultured in the permeable inserts. Five hundred microliters of the cell suspension with a density of $3 \times 10^5$ cell/mL were seeded in ThinCert cell culture inserts (transparent, PET) with a 0.4 μm pore size and a 1.131 cm$^2$ surface area (Greiner Bio-One) in a 12-well plate (TPP Tissue Culture Test Plate, TPP Techno Plastic Products AG). A basolateral compartment was filled with 1.5 mL of the complete medium. Cells were maintained at 37 °C and 5% $CO_2$. The medium was changed every two days. The development of monolayers was monitored by measuring transepithelial electrical resistance (TEER) using an EVOM3 epithelial voltmeter with an STX2-PLUS electrode (World Precision Instruments). Cell monolayers were used from 21 to 29 days after seeding. Only cell monolayers with TEER above 1000 Ω cm$^2$ were used in the transport study (Supplementary Fig. 19).

To select the concentration of 2-octynoHA for the transport experiment, cell monolayers were incubated with various concentrations of the compound from 0.1 to 1 mM with the TEER values being monitored every 30 min over 2 h. A concentration of 0.3 mM 2-octynoHA was selected due to minimally affected TEER (Supplementary Fig. 20).

The apical-to-basolateral transport assay was performed according to the previously reported protocol with modifications[68]. The monolayers were washed and then incubated in Hanks' Balanced Salt Solution (HBSS, with calcium, magnesium, no phenol red; Thermo Fisher Scientific) with 15 mM HEPES (pH 7.4) for 15–20 min. Three inserts were treated with 0.5 mL of a solution containing 0.3 mM 2-octynoHA and 5 μM lucifer yellow (LY) in buffer (HBSS, 15 mM HEPES, pH 7.4), while the basolateral compartments were filled with 1.2 mL of buffer alone. As a control, three inserts were treated with the same volume of 5 μM LY solution. The plate was incubated for 2 h on a shaker at 70 rpm (MaxQ2000 $CO_2$ Plus, Thermo Fisher Scientific). Aliquots of 0.6 mL were collected from the basolateral compartment at 0, 30, 60, 90 and 120 min and were replaced by the same volume of a buffer solution. At the end of the experiment, a sample from the apical compartment was collected followed by washing of both compartments with buffer (HBSS, 15 mM HEPES, pH 7.4). All samples were acidified using 10% (v/v) $H_3PO_4$ to a final concentration of 0.25 % (v/v) immediately after collection and were stored at −80 °C until further analysis. TEER values were measured before and after the transport experiment for all inserts while kept on the heating plate at 37 °C (Supplementary Fig. 21). The integrity of monolayers was evaluated based on the permeability of LY (Supplementary Fig. 22). LY fluorescence was measured in collected samples (ex. 428 nm, em. 536 nm) in a 96-well plate (black, flat bottom, PP; Greiner Bio-One) using a plate reader (Infinite M200, Tecan). Calculations of the apparent permeability coefficient ($P_{app}$) and mass balance were performed according to Tavelin et al.[69] and Hubatsch et al.[70].

## Quantification of 2-octynoHA in the permeability assay

Sample collected in the transport assay were analyzed using an LTQ-XL linear ion trap mass spectrometer equipped with a heated ESI II source (Thermo Scientific). The mass spectrometer was coupled to a Waters Acquity™ UPLC system (Waters Corporation). Separation was performed on a Kinetex C18 polar column (2.1 × 100 mm, 2.6 μm, 100 Å; Phenomenex) equipped with a C18 polar pre-column (Phenomenex), at 25 °C. The mobile phase consisted of 0.1% (v/v) FA in $H_2O$ (eluent A) and 0.1% FA (v/v) in ACN (eluent B). Solvents were sonicated for 15 min before the chromatographic analysis. The flow rate was set to 0.4 mL/min and the injection volume was 10 μL. The dwell volume of the UPLC system was 0.7 mL. The final LC gradient was as follows: 0–2 min at 5% B, 2–10 min to 50% B, 10–12 min to 100% B, 12–14 min at 100% B, 14–15 min to 5% B, 15–17 min at 5% B. The autosampler was set to 10 °C, respectively. Analyses of 2-octynoHA, 5-pentylisoxazol-3-ol and OHA were performed in the positive ionization mode. The ESI II source was set to 50 °C. Sheath gas and auxiliary gas were set to 34 and 11 arbitrary units, respectively: source voltage was 5.00 kV; the temperature of ion transfer capillary was 275 °C. The capillary voltage was 31 V; the tube lens voltage was 80 V. $MS^1$ was performed in full scan mode ($m/z$ 100–200). Automatic gain control was set to 15,000 ions for full scan and 5000 for $MS^n$. Collision-induced dissociation (($CID$)-$MS^n$) experiments were performed on precursor ions selected for $MS^1$ using information-dependent acquisition. $MS^2$ and $MS^3$ were performed in the IDA mode: four IDA $MS^2$ experiments were performed on the four most intensive signals from $MS^1$.

For the calibrant and QC stocks, independent solutions of 2-octynoHA and 5-pentylisoxazol-3-ol (10 mM each) were prepared in buffer (HBSS, 15 mM HEPES, pH 7.4) with 0.1% (v/v) $H_3PO_4$. From a single stock solution, the highest calibrant (Cal 1) containing both analytes (0.1 mM each) was prepared using a buffer with 0.1% (v/v) $H_3PO_4$. Cal 2–6 were obtained by diluting Cal 1 with the same buffer. QCs were prepared accordingly. $QC_{High}$ was set to 10% lower than Cal 1. $QC_{Low}$ was used at a concentration 20% higher than Cal 6. As an internal standard solution, 10 μM OHA in ACN with 0.1% (v/v) FA was used. Cal, QCs, and samples (100 μL) were diluted with 300 μL of the internal standard solution followed by centrifugation at 10,000 $x\ g$ for 10 min. The supernatant was diluted with 400 μL of eluent A and subjected to the analysis.

For the calibration curve, six calibrants in the concentration range of 2.5 to 100 μM were prepared as described above. The regression lines were calculated using a weighted [1/X$^2$] least-squares regression model. Daily calibration curves (double measurement per level) were performed. All calibration samples were compared to their theoretical value. Quantitative accuracy was given within 20% of the target.

QC samples (QC$_{Low}$ and QC$_{High}$) were analyzed in analytical duplicate on each of eight days, accordingly. Accuracy was defined as the percent deviation of the mean calculated concentration at each concentration level from the corresponding theoretical concentration. Intra-day (RSD$_R$) and inter-day precision (RSD$_T$) were calculated as relative standard deviation (Supplementary Table 12).

Autosampler stability was determined by analyzing QC$_{Low}$ and QC$_{High}$ samples from the same vial after 17 and 24 h, respectively. Peak areas of target analytes were compared to initial values. Freeze-thaw stability was tested by subjecting QC$_{Low}$ and QC$_{High}$ to four freeze-thaw cycles over a period of 15 days. After sampling, QCs were frozen again directly. Between measurements, QC samples were placed in the freezer for at least 24 h.

Carryover was tested by injecting analyte-free blank samples after the highest concentrated calibrant. Blank chromatograms were inspected for potential analyte peaks.

Peak integration was performed with Xcalibur software (Version 3.3.2 SP2). GraphPad Prism 8.2 (GraphPad Software) was used for regression analysis and 95% confidence interval (CI) determination.

## Calculation of the apparent permeability coefficient and mass balance

The apparent permeability coefficient (P$_{app}$, cm s$^{-1}$) for 5-pentylisoxazol-3-ol and LY was calculated using Eq. (2) for the sink conditions[69,70].

$$P_{app} = \frac{dQ}{dt} \cdot \frac{1}{A \cdot C_0} \qquad (2)$$

where dQ/dt is a compound permeation rate across the monolayer (μmol s$^{-1}$), A is the surface area of the insert (cm$^2$), and C$_0$ is the initial concentration of the compound (μM). In the case of 5-pentylisoxazol-3-ol, the sum of 2-octynoHA and 5-pentylisoxazol-3-ol concentrations were used.

The mass balance (recovery) for 5-pentylisoxazol-3-ol was calculated using Eq. (3)[70]:

$$Recovery\,(\%) = \left( C_{AP(2h)} \cdot V_{AP} + \sum \left( C_{S(t)} \cdot V_{S(t)} \right) \right. \\ \left. + C_{BL(2h)} \cdot V_{BL(2h)} \right) \cdot \frac{100}{C_{AP(0)} \cdot V_{AP(0)}} \qquad (3)$$

where C$_{AP}$ and C$_{BL}$ are concentrations of 5-pentylisoxazol-3-ol in the apical and basolateral compartments (μM), respectively, at the start of incubation (0) and at the end (2 h). C$_{S(t)}$ is the compound's concentration (μM) in the sample collected at different time points (t). V is the volume (cm$^3$) of the corresponding sample or compartment.

The mass balance for 5-pentylisoxazol-3-ol in the control experiment without cell monolayers and with cell monolayers were 88.2 ± 21% and 78.6 ± 9.1% (mean ± SD, n = 3), respectively.

## Pharmacokinetics of 2-octynoHA

The PK studies were conducted at the Institut National de la Recherche Scientifique (INRS) (Laval, QC, Canada). The test facility is accredited by Association for Assessment and Accreditation of Laboratory Animal Care (AAALAC) and Canadian Council on Animal Care (CCAC). All animal experiments were approved by the Comité Institutionnel de Protection des Animaux (CIPA) of the INRS (Ethical Animal Protocol #2002-04). PK of 2-octynoHA was studied in male Beagle dogs (Marshall Bioresources). Dogs were chosen over rats for the PK studies to minimize the risk of capsule retention in the stomach, which has previously been shown to occur in rats[71].

In three PK studies, a total of six animals were used. Among them, three animals were allocated to a single study (PK 1), whereas the remaining three animals were involved in two studies (PK 2 and PK 3).

Between the studies, dogs were fed twice a day (Teklad Certified 25% Lab Canine Diet, Inotiv, 8727 C) and housed in groups at 20 ± 3 °C and relative humidity of 45% (maximum range: 30-70%) with a 12 h light/dark cycle and water ad libitum. For the PK study, fasted animals were transferred to individual housing 2-3 h prior to dosing without food access. The first group (PK 1) of three animals (30–40 months old) with body weights of 10.2 ± 0.96 kg were administered I.V. with 100 mg of 2-octynoHA solubilized with HPβCD. For the administration, a solution of 75.8 mM 2-octynoHA with HPβCD (1:1 mol/mol) in ultra-pure water was filtered using a 0.2 μm syringe filter. The solution was freeze-dried in autoclaved glass vials equipped with 0.2 μm syringe filters. The lyophilized mixture was stored in sealed vials and was reconstituted in 0.9% NaCl solution (B. Braun Medical AG) before administration. Animals received a single dose via slow bolus injection into the right saphenous vein. The second group (PK 2) of three animals (36–37 months old) with body weights of 11.4 ± 0 kg was dosed orally with 300 mg of 2-octynoHA formulated in an uncoated gelatin capsule size 0 (Interdelta SA). The third group (PK 3) of three animals (39–40 months old) with body weights of 11.5 ± 0 kg received orally 300 mg of 2-octynoHA formulated in a coated gelatin capsule size 0 (Interdelta SA). For colon-specific delivery, capsules were coated with two layers of a coating solution composed of 15% (w/w) Eudragit S100, 5% (w/w) TEC, 40% (w/w) isopropyl alcohol, 40% (w/w) ethanol using a dip coating technique. Four hours post-dosing, animals were returned to their group housing and provided with food (Teklad Certified 25% Lab Canine Diet, Inotiv, 8727 C).

For PK analysis, blood samples of 5 mL were collected at predetermined time points (Supplementary Table 8) and aliquoted into two K$_2$EDTA tubes (BD Vacutainer® EDTA Tubes, BD). Samples were centrifuged for 10 min at 1700 x g at 4 °C. Plasma was collected and stored in clear polypropylene vials at −80 °C until further analysis. Quantification of 2-octynoHA in plasma samples was performed using UPLC-QTOF-MS method.

PK parameters listed in Supplementary Table 10 were calculated by non-compartmental analysis using the PKSolver 2.0 add-in[72] for Microsoft Excel. The area under plasma concentration vs. time curve was calculated by applying a linear trapezoidal rule. In PK 1 and PK 2, all PK parameters could be calculated for three dogs, except for t$_{1/2}$ and t$_{max}$ in PK 2, where these parameters could not be calculated for one dog due to the lack of data points.

Details of each of the three studies are summarized in Supplementary Table 8.

## Capsule coating preparation and disintegration study

To optimize colonic coatings for capsules administered to rats and dogs, a series of disintegration tests was performed (Supplementary Tables 6 and 7). Coating solutions composed of 13–15% (w/w) of Eudragit S100, Eudragit L100 or Eudragit L100-55 together with 5% (w/w) triethyl citrate were prepared in a mixture of ethanol and isopropanol in a 1:1 (w/w) ratio. Gelatin capsules size 0 (Interdelta SA) were coated with multiple layers (from 1 to 4) of a coating solution using a dip coating method. Coated capsules were left to dry for 2 h between layers and overnight after the last layer in the incubator at 27 °C. Coated capsules were filled with a mixture of D-mannitol and Evans Blue used as a coloring agent indicating the leakage. The ability of coated capsules to withstand physiological conditions was tested in SGF and SIF using a disintegration tester (ERWEKA). For SGF, a solution (2 L) containing 4 g NaCl was prepared, 14 mL of 12 M HCl were added to adjust pH to 1.2. For SIF, a solution (2 L) containing 13.6 g KH$_2$PO$_4$ and 1.7 g NaOH was prepared, pH was adjusted to 6.8 or 5.5 using NaOH or HCl, respectively. All capsules were first kept in SGF for 3 h at 37 °C. Then, capsules coated with Eudragit S100 solution were immersed in SIF at pH 6.8, while capsules coated with Eudragit L100 or Eudragit L100-55 solutions were immersed in 50 mM KH$_2$PO$_4$ pH 5.5, respectively. Additionally, capsules coated with 2 layers of 13–15% Eudragit

S100 or 3 layers of 13–14% Eudragit S100 were tested in 50 mM $KH_2PO_4$ pH 7.3 where they fully disintegrated within 30 min.

## Quantification of 2-octynoHA in dog plasma samples

Analysis of plasma samples was performed at the University of Montreal (Platform of Biopharmacy, Faculty of Pharmacy). Concentrations of 2-octynoHA in dog plasma after I.V. and P.O. administrations were evaluated using the Acquity UPLC I-Class system connected to the Xevo™ G2-XS QTof mass spectrometer (Waters Corporation).

Plasma samples collected after I.V. administration of 2-octynoHA were thawed on ice and then were extracted by protein precipitation prior to analysis. Forty microliters of plasma were mixed with 80 μL of an organic solvent mixture composed of ACN/MeOH (80:20 v/v) and 1% (v/v) FA. Samples were vortexed to precipitate proteins and were centrifuged for 5 min at 15,870 $x$ $g$. Subsequently, 100 μL of supernatant were collected and further diluted with 100 μL of water containing 0.1% (v/v) FA. Samples were then transferred into a 96-well sample collection plate (700 μL round well; Waters Corporation), 4 μL of plasma extract was injected into the UPLC system. Chromatographic separation was performed on an Acquity UPLC BEH C18 column (2.1 mm × 50 mm, 1.7 μm, 130 Å) (Waters Corporation) maintained at 50 °C using a gradient-elution method at a flow rate of 0.4 mL/min. A following gradient of A ($H_2O$ + 0.2% (v/v) FA) and B (ACN + 0.2% (v/v) FA) was applied: 0.3 min at 7% B, from 7 to 55% B in 5 min, from 55 to 98% B in 0.2 min, 1 min at 98% B, followed by equilibration at 3% B for 1.8 min. Spectra were acquired in a positive ion mode with the following instrument settings: source temperature 120 °C, desolvation temperature 550 °C, capillary voltage 2.5 kV, sample cone voltage 30 V, cone gas flow 80 L/h, desolvation gas flow 1000 L/h. The $m/z$ scan range was set from 50 to 800, samples were analyzed using a TOF-MS$^E$ scan mode with a scan time of 0.15 s. The data was processed using UNIFI 1.9.4 software (Waters Corporation).

The calibration curve was prepared in duplicate. A stock solution of 2-octynoHA was prepared at 5 mg/mL in MeOH/$H_2O$ (1:1 v/v) solvent mixture containing 0.1% (v/v) FA. Intermediate standard solutions were prepared at 40 μg/mL and 100 μg/mL using the same acidified solvent mixture. Dog plasma was spiked with appropriate volumes of the 40 μg/mL and 100 μg/mL solutions to a final volume of 200 μL to generate standards at 20, 50, 200, 500, 2000 and 5000 ng/mL. The standards were then processed as described above. The calibration curve was fitted using the quadratic regression with a weighing factor of 1/X. The LLOQ was defined as the lowest calibrator within ± 30% of the nominal concentration.

Frozen plasma samples collected after P.O. administration of 2-octynoHA were treated with $H_3PO_4$ to a final concentration of ca. 1% (v/v). The samples were vortexed periodically every 5 min until completely thawed and then extracted by protein precipitation. In brief, 40 μL of plasma were precipitated with 80 μL of organic solvent consisting of ACN/MeOH (80:20 v/v) and 0.2% (v/v) FA. To precipitate proteins, samples were vortexed and centrifuged for 5 min at 15,870 $x$ $g$. Hundred microliters aliquot of the supernatant was further diluted with 100 μL of water containing 0.1% (v/v) FA. The sample extracts were transferred into a 96-well plate (450 μL well, Thermo Fischer Scientific), 5 μL was injected into the UPLC system equipped with Acquity UPLC BEH C18 column (2.1 mm × 100 mm, 1.7 μm, 130 Å) (Waters Corporation). The column temperature was kept at 40 °C. Sample separation was performed at a flow rate of 0.4 mL/min using the following gradient of A (water + 0.2% (v/v) FA) and B (ACN + 0.2% (v/v) FA): 0.9 min at 3% (v/v) B, from 3 to 50% (v/v) of B in 6 min, from 50 to 98% (v/v) of B in 1.3 min, 0.7 min at 98% (v/v) B, followed by equilibration at 3% (v/v) B for 2.8 min. Positive ion mode masses were scanned from 50 to 900 $m/z$ using a TOF-MS$^E$ scan mode with a scan time of 0.12 s. The following source parameters were utilized: source temperature 120 °C, desolvation temperature 550 °C, capillary voltage 2.0 kV, sample cone voltage 30 V, cone gas flow 80 L/h, desolvation gas

flow 1000 L/h. All acquired data were processed using Waters Connect 2.2 with UNIFI 2.1.2.14 software (Waters Corporation).

For the calibration curve, a stock solution of 5 mg/mL 2-octynoHA was prepared in MeOH/$H_2O$ (20:80 v/v) mixture containing 0.1% (v/v) FA. Intermediate standard solutions at 40 μg/mL and 100 μg/mL were prepared from the stock. Dog plasma was first acidified by adding phosphoric acid to a final concentration of 1% (v/v) and then was spiked with an appropriate volume of 40 μg/mL and 100 μg/mL solutions to a final volume of 200 μL to obtain standards at 20, 50, 100, 200, 500, 1000 ng/mL. Standards were processed in the same way as samples collected after P.O. administration. Two measurements of the calibration curve were performed. The quadratic regression with a weighing factor of 1/X was used to fit the calibration curve. The LLOQ was defined as the lowest calibrator within ± 30% of the nominal concentration.

## In vivo study in the acute liver injury rat model

The study was performed by Wuhan Servicebio Technology Co., Ltd. (Wuhan, China). All animal experiments were approved by the Institutional Animal Care and Use Committee (IUCAC) of Wuhan Servicebio Technology Co., Ltd. (protocol numbers: 2021071, 2021136, 2021137, 2022003). Male Sprague Dawley rats (6-8 weeks old) were obtained from the Beijing Vital River Laboratory Animal Technology Co., Ltd. Animals were housed in cages (3-4 rats per cage) in the animal facility at 22-23 °C with humidity within a range of 40-70%, 12 h light/dark cycle, and free access to water and food (Lab Rat and Mouse Maintenance Diet, Jiangsu Xietong Pharmaceutical Bio-engineering Co., Ltd., 1010086).

To induce liver damage, four groups of 10 animals each (40 animals in total) were subjected to an I.P. injection of N-nitrosodiethylamine (DEN) solution at a dose of 60 mg/kg on days 1, 3, 5, and 7 during the 8-day study. The first group of animals was treated with an HPβCD aqueous solution at a dose of 270 mg/kg. The second and third groups were treated with rifaximin suspension in ultra-pure water at 30 or 60 mg/kg, respectively. The fourth group was treated with a solution of 2-octynoHA and HPβCD (1:1 mol/mol) at a dose of 15 mg/kg and 135 mg/kg, respectively. All treatments were administered by oral gavage twice a day from days 3 to 8 of the study. To monitor blood ammonia levels, blood was sampled sublingually a day before the first DEN injection (day 0), and on days 6 and 8 before the evening administration of the treatment. During blood sampling, rats were kept at 2–2.5% isoflurane anesthesia. Blood ammonia concentration was measured immediately upon blood withdrawal using an ammonia blood meter PocketChem BA PA-4140 (Arkray) following the manufacturer's instructions. Ammonia concentrations below the measurement range of the instrument (7–286 μmol/L) were set equal to 7 μmol/L. The body weight of rats was monitored every day. At the end of the study, all rats were subjected to carbon dioxide euthanasia.

## Preparation of 2-octynoHA formulations for the in vivo study in BDL rats

A suspension of 2-octynoHA was prepared in an aqueous solution of 1% (w/v) HPMC and 0.004–0.008% (w/v) neotame. 2-octynoHA was gradually mixed with a vehicle solution and vortexed just before administration to a final concentration of 30 mg/mL.

A solution of 2-octynoHA was prepared by adding an aqueous solution of HPβCD at a 1:1 mol/mol ratio. To facilitate solubilization, the mixture was incubated at 37 °C with shaking. The solution was lyophilized and the product was stored at 4 °C. Prior to administration, the lyophilized product was reconstituted by adding ultra-pure water to a final concentration of 80 mM.

Due to the slight acidity of the GI tract of rats[47], Eudragit L100 was used to create a capsule coating for colonic delivery. A mixture of 15% (w/w) Eudragit L100, 5% (w/w) TEC, 40% (w/w) isopropyl alcohol, 40% (w/w) ethanol was prepared and left to stir until a clear solution was obtained. Caps and bodies of gelatin

capsules size 9 h (Torpac Inc.) were first coated with one layer of the coating solution using a dip coating method and were left to dry for 2 h at RT. Caps were then coated with a second layer in the same manner. Both caps and bodies were dried overnight at RT. Bodies were filled with ca. 10 mg of 2-octynoHA on the balance using a funnel, a tamper and a stand (Torpac Inc.). Bodies of closed capsules were coated with the same solution by immersing them only up to the cap to ensure sealing. Suspended capsules were dried overnight at RT and then stored in glass vials at 4 °C until the administration.

### In vivo study in BDL rats

All animal experiments were conducted according to the Federal and local ethical guidelines, the protocols were approved by the local Committee on Animal Experimentation for the Canton de Vaud, Switzerland (VD3022.1). Male Wistar rats were obtained from Charles River Laboratories (L'Arbresle, France) at 150–175 g of weight. Animals were housed in cages (2-3 rats per cage) in the animal facility at 20–24 °C with a 12 h light/dark cycle, and free access to water and food (SAFE 150 SP-25, SAFE, U8409G10R).

To induce type C HE, rats underwent bile duct ligation surgery[51] according to the protocol described in detail in reference[73]. The study included 55 animals allocated to four groups. The first group (n = 12) was administered a suspension of 2-octynoHA at a dose of 30 mg/kg in a solution containing 1% (w/v) HPMC and 0.004–0.008% (w/v) neotame twice a day (suspension group). The second group (n = 10) was administered a solution of 2-octynoHA at a dose of 30 mg/kg twice a day (solution group). The third group of animals (n = 20) was administered a coated capsule with 10 mg of 2-octynoHA (ca. 30 mg/kg) once a day (capsule group). The control group received the solution containing 1% (w/v) HPMC and 0.004-0.008% (w/v) neotame (n = 13) (negative control group). All treatments were administered via oral gavage or voluntarily consumed (negative control and suspension groups, as the animals were trained to take the treatment voluntarily) starting from day 35 post-surgery and were continued for 5 days until day 39. During the study, animals were weighed every 2–3 days. In order to reduce the risk of ammonia contamination, blood for ammonia and bilirubin measurements was sampled sublingually[52,53,59]. Blood samples were collected before the surgery (day 0), 2 weeks post-surgery (day 14), before the treatment (day 35), in the middle of the treatment (day 37) and at the end of the treatment (day 39). For the sampling, rats were kept for 4 min at 4% isoflurane, 0.5 L/min air and 0.5 L/min oxygen during induction, then they were kept for 1 min at 2–3% isoflurane during sampling[74]. Blood ammonia concentrations were determined in whole blood samples immediately after collection using a blood ammonia meter PocketChem BA PA-4140 (Arkray). For bilirubin and glucose, blood was collected in pre-cooled $K_3$EDTA tubes (Sarstedt AG & Co. KG), and centrifuged at 2700 x g for 7 min at 4 °C. Bilirubin and glucose concentrations were measured in plasma using the Reflotron Plus system (Roche Diagnostics). After the treatment on day 39, in vivo $^1H$ MRS measurements of brain metabolites (Asc, tCho, Cr, PCr, GABA, Gln, Glu, Ins, Lac, NAA, NAAG, PE, Tau and Glc) were performed to evaluate the effects of the treatments on the neurometabolic profile of BDL rats. At the final point of the MRS experiment, animals were euthanized by decapitation under anaesthesia without awakening.

### In vivo $^1H$ MRS in BDL rats

Animals were subjected to $^1H$ MRS scan on day 39 post-surgery. During the MRS experiments, the rats were kept under 1.5–2% isoflurane anaesthesia with respiration rate maintained at 60–70 breaths/min and body temperature at 37.5–38.5 °C. $^1H$ MRS measurements were conducted on a horizontal actively shielded 9.4 Tesla system (Magnex Scientific) interfaced to a Varian Direct Drive console using a home-built quadrature surface coil as a transceiver (17 mm diameter for each loop). The volume of interest (VOI = 2.5 × 2.5 × 2.5 mm³) for the $^1H$ MRS

scans was placed in the cerebellum, and spectra were acquired using the ultra-short-echo time SPECIAL sequence with the echo time (TE) = 2.8 ms, repetition time (TR) = 4 s and 160 averages[75]. First and second-order shims were adjusted using FASTMAP[76] reaching water resonance linewidths 13–16 Hz. The VOI was localized on axial and sagittal anatomical $T_2$ weighted images (multislice turbo-spin-echo sequence, $(TR/TE_{eff})$ = 4000/52 ms, field of view = 23 × 23 mm², 256 × 256 image matrix). Outer volume suppression (OVS) was used to improve signal localization in combination with VAPOR water suppression[77].

Spectra were fitted and metabolite concentrations were calculated by LCModel[78] and expressed in mmol/kg$_{ww}$ using the unsuppressed water signal from the same VOI as an internal reference. Only metabolites with CRLB lower than 30 % were considered for further analysis. The ultra-short echo-time MRS allowed the detection of 14 metabolites, all included in basis-set: Asc, tCho, Cr, PCr, GABA, Gln, Glu, Ins, Lac, NAA, NAAG, PE, Tau and Glc. tCr was calculated as a sum of Cr and PCr. In addition to the in vitro measured metabolites, the basis-set for spectral fitting also contained in vivo acquired macromolecule spectrum[79,80]. Only the metabolites showing a response to the disease are presented.

### Statistical analysis

All data are presented as mean ± SD from $n$ experiments as indicated in each figure legend. The statistical analysis was performed using GraphPad Prism version 9.4.1 for Windows (GraphPad Software). In the cytotoxicity experiments, statistical significance within the treatment group was calculated by one-way ANOVA with Tukey's multiple comparisons test. In animal studies, statistical significance between the treatment groups was calculated by two-way repeated measures ANOVA with Tukey's multiple comparisons test, while statistical significance between treatment days within one treatment group was calculated by one-way repeated measures ANOVA with Tukey's multiple comparisons test (e.g. ammonia, bilirubin, glucose and body weight in DEN and BDL rats). For the brain metabolites in BDL rats, statistical significance between the treatment groups was calculated by one-way ANOVA with Tukey's multiple comparisons test. A $p$- value < 0.05 was considered statistically significant.

### Reporting summary

Further information on research design is available in the Nature Portfolio Reporting Summary linked to this article.

## Data availability

The urease structures described and visualized in the Introduction section were obtained from the Protein Data Bank (PDB) with accession codes 4H9M, 4UBP, 1E9Y, 1FWE, 6ZJA. The data generated in the in vitro studies (caecal content-based and pH-based cell-free urease activity assays, cytotoxicity, mutagenicity, stability, permeability experiments), PK and in vivo studies in rats are provided in the Source Data file. Source data are provided with this paper. Results of statistical analysis are provided in the Supplementary Data file. Raw spectra data generated in $^1H$ MRS experiments are available in Zenodo (https://doi.org/10.5281/zenodo.8042020). X-ray crystallographic data for 5-pentylisoxazol-3-ol have been deposited at the Cambridge Crystallographic Data Centre under deposition number 2257012 (https://dx.doi.org/1s0.5517/ccdc.csd.cc2frlvn). The remaining data generated in this study are available in the Supplementary Information. Source data are provided with this paper.

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

## Acknowledgements

This work was supported by a grant from Carigest SA to J.-C.L. The authors further thank M. D. Wörle, M. Solar, and N. Trapp (Small Molecule Crystallography Center, ETH Zurich) for the single-crystal x-ray crystallographic analysis, Susanne Freedrich (ETH Phenomics Center) for providing samples of caecal content, Suiying Ye (ETH Zurich) for her help with organizing in vivo studies. The [1]H MRS experiments were made possible thanks to the Center for Biomedical Imaging (CIBM) of the Lausanne University Hospital (CHUV), University of Lausanne (UNIL), Ecole Polytechnique Fédérale de Lausanne (EPFL), University of Geneva (UNIGE), Geneva University Hospitals (HUG), the Leenaards and Jeantet Foundations. Figures 4d and f were created with BioRender.com.

## Author contributions

D.E.: design, execution and analysis of in vitro studies (i.e. urease activity, cytotoxicity, mutagenicity, stability, permeability and capsule disintegration experiments), preparation of formulations for in vivo studies, analysis of in vivo results, manuscript writing and revision. F.I.: design, synthesis and characterization of HAs, manuscript revision. Y.B.: synthesis and characterization of HAs, manuscript revision. Z.L.: assistance

with design and planning of in vivo studies in a rat model of acute liver injury, manuscript revision. B.A.: design, synthesis and characterization of HAs, manuscript revision. S.K.: execution and analysis of in vitro urease activity tests for NHA and DHA, manuscript revision. C.S.: development of the LC-MS method, analysis of samples collected in the permeability study, manuscript writing and revision. E.M.: assistance with the development of LC-UV/Vis method and sample analysis, manuscript revision. T.C.: support in the discussion of the structure-activity relationship of HAs, manuscript revision. D.Simicic: design, execution and analysis of in vivo studies in BDL rats and 1H MRS experiments, manuscript writing and revision. D.Sessa: execution of BDL surgery, treatment administration, sample preparation and analysis, manuscript revision. S.-O.M: execution of BDL surgery, treatment administration, sample preparation and analysis, manuscript revision. K.P.: design of in vivo studies in BDL rats, sample preparation and analysis, manuscript revision. C.R.C.: design, coordination, execution and interpretation of in vivo studies in BDL rats and $^1$H MRS experiments, manuscript writing and revision. C.E.A.: assistance with the pH-based urease activity study design and manuscript revision. J.-C.L.: overall conception, overall design, overall coordination, overall interpretation, manuscript writing and revision.

## Funding

## Competing interests

J.-C.L., D.E., and F.I. are co-inventors on a patent application (WO 2023/275790 A1) related to the technology described in this manuscript. This patent application has been licensed to Versantis AG, a GENFIT Group company. J.-C.L. is a consultant for GENFIT. D.E. is an employee of GENFIT. The remaining authors declare no competing interests.
