## [Peer Review File · Nature Communications]

REVIEWER COMMENTS

Reviewer #1 (Remarks to the Author):

The manuscript describes a study to identify a series of hydroxamate-based inhibitors of urease for the treatment of hepatic encephalopathy, among which they identify 2-octynohydroxamic acid (2-octynoHA) as the best candidate. My competence does not cover the cytotoxicity nor the pharmacokinetic sections of the study, nor the in vivo assays, so my comments will be limited to the chemistry and structural studies.

The first evident flaw of the paper is the failure to report and comment on the existence of four crystal structures of different ureases inhibited by hydroxamates (PDB codes 4H9M, 4UBP, 1E9Y and 1FWE). If the authors had focused only on the pharmaceutical aspects of the proposed new drug, it would have been more acceptable. However, the fact that they tried to understand the binding mechanism of 2-octynoHA using molecular dynamics (paragraph 2.3) without ever citing the known structures of urease-hydroxamate, reveals that the authors have not done their homework. Furthermore, the authors failed to compare their best binding pose with the experimental structures, a key step to assess the reliability of the reported calculations.

Additional comments: the identified best inhibitor does not appear to be stable in solution, as it undergoes a cyclisation reaction with the loss of the hydroxamate functionality. The authors appear to indicate that this is a positive aspect of the potential drug, as it has been reported that the presence of a hydroxamate functional group leads to cytotoxicity and mutagenicity; however, considering that this group is essential for urease inhibition, it does not make much sense that the molecule is proposed as a drug to treat urease-induced pathologies, considering its instability.

For the above reasons, I do not support publication of this study.

Reviewer #2 (Remarks to the Author):

The paper describes novel hydroxamate derivatives for treating hepatic encephalopathy, a disease characterized by elevated amounts of ammonia in the blood produced by the action of colonic bacterial urease on urea. The strength of the article i) is the production of new compounds that are more active than the reference drug, an antibiotic named Rifamixine, used in the same therapeutic application ii) addressing a disease with real medical needs.

Several molecules were designed, among which a novel compound 2-octyl hydroxamic acid, demonstrated a powerful urease-inhibiting effect in vitro. One control is missing at this level: cyclodextrins (CD) are employed to solubilize the compounds. CD might interfere with enzyme inhibition, although the data in vivo related to its influence are unclear. Moreover, molecular dynamics simulations did not confirm the higher binding efficiency of 2-octyl hydroxamic, and the question remains whether the cyclodextrins have to do with the inhibitory observed in vitro. CD was not included as a control in some experiments such as cytotoxicity. Of course, in vivo, the negative control made of CD has a lower effect on reducing blood ammonia in the DEN model after 6 days. However, without another control consisting of untreated animals, it is hard to understand if it does not contribute to any effect.

In the DEN model, 2-octyl hydroxamic was slightly superior to Rifaximin at 60 mg/Kg. It is unclear why this drug reference was not included in all further studies, particularly in the BDL model.

Why was pharmacokinetics carried out in dogs and efficacy in rat models?

Why were all delivery systems not tested for PK in the rat model?

Regarding the brain metabolites, it is surprising that the solution is more affecting the reduction of metabolites than the suspension or enteric capsule. A better understanding of the process involved in these differences is needed.

In conclusion, the paper contains overwhelming data with clear evidence that 2-octyl hydroxamic could be a novel promising drug in treating hepatic encephalopathy.

Reviewer #3 (Remarks to the Author):

The manuscript by Evstafeva et al describes a very complete study on the selection of a compound optimized to inhibit urease from bacteria in the intestine so as to decrease hyperammonemia in vivo and thus be able to be used as a treatment for HE.

The study is very complete, including the chemical selection of the best inhibitor, testing its best formulation, pharmacokinetics and evaluating in vivo efficacy in an acute and a chronic model of liver failure and HE. In addition, they analyze some metabolic consequences at the brain level.

Here are some criticisms and suggestions about the details of the work:

1. The title could be more informative, with more specific details about the results and conclusions of the study and not so general, that it is not really known what the work is about, reading only the title.
2. In the introduction, the following statement "The fact that rifaximin shows some activity in HE supports the importance of ammoniogenesis by the gut microbiota" is not entirely correct since rifaximin not only decreases ammonium levels but its beneficial effect on HE may be due to different causes, since it alters the microbiota and its metabolism in general, altering also the immune system and different metabolites in addition to ammonium, such as bile acids, tryptophan, short-chain fatty acids, etc. Better remove this sentence.
3. Although it depends on the journal's instructions, in the introduction it is not necessary (better not) to write the results of the study, only the objectives and approach to develop them.
4. The authors could justify the use of human colorectal adenocarcinoma cells (Caco-2) to assess cell viability, toxicity of the compound. I'm not sure these cells are a good model. It would be better to use unmodified cells, from the intestinal epithelium, for example.
5. It would be necessary to justify the use of the model of acute liver damage due to the administration of DEN, since it is not one of the more used models of HE (See DeMorrow et al. 2021 ISHEN guidelines on animal models of hepatic encephalopathy *Liver Int.* 2021 Jul;41(7):1474-1488. doi: 10.1111/liv.14911).

6. It would also be necessary to better discuss why the effect on ammonium levels disappears on day 8, it is transitory, in the acute liver failure model. What relevance can this have in patients with HE?

7. Why is ammonium measured in blood drawn sublingually? Give a reference and justify why this method is chosen. Also, daily anaesthesia with isoflurane could affect the results, this should be discussed.

8. In the HE model of chronic liver failure due to biliary ligation, the treatment time may be too late. In this model, the rats 5 weeks after surgery are severely affected and begin to die. It would be better to try the treatment a week or two before. The decrease in hyperammonemia in these conditions seems very slight, and, although significant in some cases, little relevant.

9. The effects of treatments, or lack of effect, on bilirubin have not been discussed.

10. A relevant problem is the lack of controls, sham-operated rats, to observe the alterations present in BDL rats, mainly in the results of metabolite levels in the cerebellum.

11. Why has the cerebellum been chosen to analyze changes in brain metabolites? Authors must justify it and comment that the changes can be different in different brain regions.

12. The fact that all metabolites are decreased by treatments, specially solution, seems to be a general effect, rather than specific to particular metabolites, when they are as different as glucose or the GABA neurotransmitter.

In summary, the work deals with a complete study on the optimization of the best urease inhibitor and the best formulation to be able to use it as a treatment to reduce hyperammonemia, which could improve HE. In addition, the gut absorption, pharmacokinetics, stability and effect of its main metabolite are evaluated.

However, the results in the in vivo models used are not very conclusive and it does not seem that the chosen inhibitor has very significant effects on hyperammonemia, especially in the chronic liver failure model, compared to other treatments previously used to reduce hyperammonemia in HE.

This should be better discussed and the study expanded with better models.

Reviewer #1

The manuscript describes a study to identify a series of hydroxamate-based inhibitors of urease for the treatment of hepatic encephalopathy, among which they identify 2-octynohydroxamic acid (2-octynoHA) as the best candidate. My competence does not cover the cytotoxicity nor the pharmacokinetic sections of the study, nor the *in vivo* assays, so my comments will be limited to the chemistry and structural studies. The first evident flaw of the paper is the failure to report and comment on the existence of four crystal structures of different ureases inhibited by hydroxamates (PDB codes 4H9M, 4UBP, 1E9Y and 1FWE). If the authors had focused only on the pharmaceutical aspects of the proposed new drug, it would have been more acceptable. However, the fact that they tried to understand the binding mechanism of 2-octynoHA using molecular dynamics (paragraph 2.3) without ever citing the known structures of urease-hydroxamate, reveals that the authors have not done their homework. Furthermore, the authors failed to compare their best binding pose with the experimental structures, a key step to assess the reliability of the reported calculations.

We thank the reviewer for this comment and agree that the simulation data were a weak point of the manuscript. As was advised, we decided to narrow our focus to the pharmaceutical aspects of 2-octynoHA and thus, omit molecular dynamics (MD) simulations data, which represented a very small fraction of the content. The primary reason for this decision is based on our intention to maintain a more specific and cohesive discussion on the therapeutic potential of 2-octynoHA which has been explored in the *in vitro* and *in vivo* studies. Therefore, we believe that by removing this part, we could highlight the most important findings on the pharmaceutical attributes of our lead compound as well as improve the clarity of the overall manuscript.

Additional comments: the identified best inhibitor does not appear to be stable in solution, as it undergoes a cyclisation reaction with the loss of the hydroxamate functionality. The authors appear to indicate that this is a positive aspect of the potential drug, as it has been reported that the presence of a hydroxamate functional group leads to cytotoxicity and mutagenicity; however, considering that this group is essential for urease inhibition, it does not make much sense that the molecule is proposed as a drug to treat urease-induced pathologies, considering its instability. For the above reasons, I do not support publication of this study.

As a hydroxamate group was previously linked to mutagenic and cytotoxic properties of some of the HAs,¹ we hypothesized that a HA-based urease inhibitor whose stability can be modulated would represent a promising candidate as it might exhibit the anti-ureolytic activity for a limited period of time and then lose a hydroxamate group minimizing the risk of adverse effects associated with this functional group.

Our study showed that acidification of the medium can significantly slow down the conversion of 2-octynoHA into 5-pentylisoxazol-3-ol (Fig. 2e). These results suggest that acidifying excipients can be used, for example, in the capsule formulation of 2-octynoHA to create an optimal environment for the compound upon dissolution and thus to prolong its stability and anti-ureolytic activity. Indeed, the addition of pH-adjusting excipients to the drug formulation in order to improve its stability is a common practice in formulation development.² Despite this, we cannot exclude

that that the currently identified lead structure may not be stable enough in clinical trials and that its structure would have to be tuned to improve its half-life.

Reviewer #2

The paper describes novel hydroxamate derivatives for treating hepatic encephalopathy, a disease characterized by elevated amounts of ammonia in the blood produced by the action of colonic bacterial urease on urea. The strength of the article i) is the production of new compounds that are more active than the reference drug, an antibiotic named Rifamixine, used in the same therapeutic application ii) addressing a disease with real medical needs. Several molecules were designed, among which a novel compound 2-octyl hydroxamic acid, demonstrated a powerful urease-inhibiting effect in vitro. One control is missing at this level: cyclodextrins (CD) are employed to solubilize the compounds. CD might interfere with enzyme inhibition, although the data in vivo related to its influence are unclear. Moreover, molecular dynamics simulations did not confirm the higher binding efficiency of 2-octyl hydroxamic, and the question remains whether the cyclodextrins have to do with the inhibitory observed in vitro.

As suggested, we included the missing data on the effect of (2-hydroxypropyl)- β -cyclodextrin (HP β CD) on the urease activity in the caecal content assay (Supplementary Fig. 1). Additionally, we assessed the urease inhibitory activity of HP β CD in a pH-based assay with Jack bean urease according to the method described in the Methods section 4.4. In brief, 800 mM HP β CD stock solution was prepared in ultra-pure water using sonication and heating at 37°C. In a 96-well plate, a mixture of 0.01 mg/mL phenol red, 100 mM urea, HP β CD (0 – 80 mM) in 15 mM KH₂PO₄ (pH 6.8) was prepared and supplemented with urease (0.1 mg/mL or 2.6 U/mL) to initiate a reaction of urea hydrolysis. The absorbance was monitored at 560 nm for 2 h, acquired spectra were analyzed as described in the Methods section 4.4. As shown in **Figs. R1 and R2** below, minimal changes in phenol red absorbance and hence in pH and ammonia production were observed in the presence of HP β CD compared to negative control.

Fig. R1. Jack bean urease activity in the presence of HPβCD in a pH-based assay with phenol red. (a) The OD₅₆₀ values of samples containing urease, urea and phenol red with increasing concentrations of HPβCD (0 – 80 mM). (b) The OD₅₆₀ values of control samples (without urease) containing urea and phenol red with increasing concentrations of HPβCD (0 – 80 mM). The OD₅₆₀ was monitored every 60 s over 2 h at 37 °C. Data are represented as lines connecting mean values from three experiments, the area within the standard deviation (SD) is shaded.

Fig. R2. *In vitro* anti-ureolytic activities of HPβCD toward Jack bean urease. T₅₀ is the time when the absorbance value of a sample with HPβCD equals to half of the maximum absorbance value of a corresponding control sample without HPβCD ($A_{1/2}$). Data are expressed as mean \pm SD from three experiments.

CD was not included as a control in some experiments such as cytotoxicity.

For the cell viability assay, HAs (AHA, OHA, 2-octynoHA) were dissolved directly in the cell medium without HPβCD. Therefore, HPβCD was not included as a control in the cytotoxicity assessment.

It has to be mentioned that it was possible to dissolve all three compounds at 10 mM concentration in aqueous media without additional solubilizing agents. Despite this, for the *in vitro* assay in

caecal content samples, HP β CD was used to prepare solutions of almost all tested HAs for the following reasons. First of all, stock solutions of some saturated HAs could not be prepared without a solubilizing agent, therefore, in order to have comparable conditions across the series of tests with different HAs, HP β CD was also used to prepare lower concentration stock solutions of other HAs (e.g. unsaturated derivatives of OHA) as well. Secondly, we anticipated that for *in vivo* studies, highly concentrated solutions of the lead compound would be required for oral administration, therefore, the effect of 2-octynoHA formulation with HP β CD on urease activity was already assessed at the stage of the *in vitro* screening.

Of course, *in vivo*, the negative control made of CD has a lower effect on reducing blood ammonia in the DEN model after 6 days. However, without another control consisting of untreated animals, it is hard to understand if it does not contribute to any effect.

We understand this criticism but considering the lack of effect of HP β CD on the urease activity *in vitro* as well as our efforts to minimize the number of animals used in experiments in accordance with the 3Rs principle, we did not include a group receiving no HP β CD in the DEN-induced liver disease model. In any case, this additional control would not change the conclusion about the activity of the urease inhibitor.

In the DEN model, 2-octyl hydroxamic was slightly superior to Rifaximin at 60 mg/Kg. It is unclear why this drug reference was not included in all further studies, particularly in the BDL model.

A lack of effect of rifaximin on hyperammonemia in the BDL rat model was previously shown by some research groups including co-authors of this manuscript.^{3,4} Since ammonemia was one of the main endpoints in our studies in the BDL model, it was decided to not include rifaximin as a reference drug to reduce the animal burden.

Why was pharmacokinetics carried out in dogs and efficacy in rat models?

Dogs were chosen to assess the PK of 2-octynoHA following oral administration of the uncoated and coated capsules instead of rats because it has been previously shown that in rats coated capsules might be retained for a long time in the stomach and even fail to exit it.⁵ Indeed, Gómez-Lado et al. studied the gastric emptying of enteric coated capsules of sizes 9 and 9h in Sprague-Dawley rats. They showed that the majority of capsules size 9 was retained for over 6 h and then disintegrated in the stomach. The gastric emptying of capsules size 9h was faster than size 9 likely due to the smaller size, nevertheless, in half of the animals these capsules reached the intestine only in 3 h after administration, and in one rat out of six rats, a capsule retained in the stomach for 5 h. Thus, to minimize the risk of capsule retention in the stomach which can alter the capsule disintegration profile and compromise the PK results, we decided to conduct these studies in dogs. The corresponding discussion and the reference have been added in the Methods section 4.10 (p. 24-25, lines 555-557).

Why were all delivery systems not tested for PK in the rat model?

This question is addressed in the previous comment.

Regarding the brain metabolites, it is surprising that the solution is more affecting the reduction of metabolites than the suspension or enteric capsule. A better understanding of the process involved in these differences is needed.

We assume that the effect of the treatment with 2-octynoHA solution in reducing ammonemia and brain glutamine was stronger than that of a suspension and enteric capsule because the compound was solubilized and hence reached adequate concentrations in the colon to inhibit bacterial ureases and hence ammonia production, although some amounts of the compound might have been absorbed in the small intestine. Since glutamine (Gln) in astrocytes is produced from glutamate (Glu) and ammonia, reduced blood ammonia levels could have led to decreased Gln production in the brain.

As for the other brain metabolites, it is unclear what could be the reason for lower taurine (Tau), creatine (Cr), phosphocreatine (PCr), total creatine (tCr), glutamate (Glu), GABA, glucose (Glc) and NAA in the 2-octynoHA solution vs. negative control group. Nevertheless, possible explanations have been provided in the main text (p. 14-15, lines 312-326).

In conclusion, the paper contains overwhelming data with clear evidence that 2-octyl hydroxamic could be a novel promising drug in treating hepatic encephalopathy.

We thank the reviewer for the overall positive assessment of our work.

Reviewer #3

The manuscript by Evstafeva et al describes a very complete study on the selection of a compound optimized to inhibit urease from bacteria in the intestine so as to decrease hyperammonemia in vivo and thus be able to be used as a treatment for HE. The study is very complete, including the chemical selection of the best inhibitor, testing its best formulation, pharmacokinetics and evaluating in vivo efficacy in an acute and a chronic model of liver failure and HE. In addition, they analyze some metabolic consequences at the brain level. Here are some criticisms and suggestions about the details of the work:

1. The title could be more informative, with more specific details about the results and conclusions of the study and not so general, that it is not really known what the work is about, reading only the title.

As suggested by the reviewer, we changed the title to "Inhibition of urease-mediated ammonia production by 2-octynohydroxamic acid in hepatic encephalopathy".

2. In the introduction, the following statement "The fact that rifaximin shows some activity in HE supports the importance of ammoniogenesis by the gut microbiota" is not entirely correct since rifaximin not only decreases ammonium levels but its beneficial effect on HE may be due to different causes, since it alters the microbiota and its metabolism in general, altering also the immune system and different metabolites in addition to ammonium, such as bile acids, tryptophan, short-chain fatty acids, etc. Better remove this sentence.

As suggested by the reviewer, we removed this sentence.

3. Although it depends on the journal's instructions, in the introduction it is not necessary (better not) to write the results of the study, only the objectives and approach to develop them.

As suggested by the reviewer, we have modified the introduction and removed the description of the results of the study (p. 5, lines 80-87).

4. The authors could justify the use of human colorectal adenocarcinoma cells (Caco-2) to assess cell viability, toxicity of the compound. I'm not sure these cells are a good model. It would be better to use unmodified cells, from the intestinal epithelium, for example.

The Caco-2 cell line is an established and commonly used model for various assays including drug absorption and cell viability testing.⁶ Although Caco-2 cells might not fully reflect the characteristics of intestinal epithelial cells, we believe that at the early stage of the project, utilization of modified cells was more time- and cost-efficient compared to the primary cells. A note on the use of Caco-2 cells for cell viability test was added in the main text (p. 8-9, lines 163-169).

5. It would be necessary to justify the use of the model of acute liver damage due to the administration of DEN, since it is not one of the more used models of HE (See DeMorrow et al. 2021 ISHEN guidelines on animal models of hepatic encephalopathy Liver Int. 2021 Jul;41(7):1474-1488. doi: 10.1111/liv.14911).

As the reviewer noted, the DEN-induced liver disease model is not one of the recommended models of HE associated with acute liver failure. In fact, DEN is a hepatotoxin and hepatocarcinogen, which is used to induce a liver cancer model.^{7,8} However, to assess the ability of 2-octynoHA to reduce systemic ammonia, we were interested in a liver disease model that can generate high ammonia levels regardless of the etiology of the underlying liver disease.⁹ Thus, we assumed that repeated administration of high doses of hepatotoxin may cause liver damage and reduce its function leading to decreased capacity to metabolize ammonia and thereby to hyperammonemia. A corresponding justification for the use of this model was added to the main text (p. 12, lines 250-255).

In addition, we appreciate the reviewer's reference to the 2021 ISHEN guidelines for animal models of HE. This reference has been included in the discussion of the BDL model, as it is one of the recommended models for type C HE (p. 13, line 272).

6. It would also be necessary to better discuss why the effect on ammonium levels disappears on day 8, it is transitory, in the acute liver failure model. What relevance can this have in patients with HE?

We assume that the blood ammonia levels on day 8 in the DEN model were comparably high among all treatment groups including rifaximin and 2-octynoHA because at this stage rats were severely affected by repeated injections of the hepatotoxin and likely experienced end-stage liver disease. Thus, regardless of the treatment that reduces ammonia production *via* some of the pathways (e.g. bacterial metabolism), a failed liver function might have led to a situation where the rest of ammonia which is still being produced through other pathways (e.g. Gln deamination) is metabolized less efficiently than on day 6 causing the elevation of ammonia in all of the

groups. In this context, sole treatment with a urease inhibitor might not be sufficient. The corresponding discussion was expanded in the main text of the manuscript (p. 13, lines 266-269).

Based on the results of the *in vivo* studies in the DEN and BDL models, we think that a combination of treatments targeting different sources of ammonia in the body might be more efficacious in late-stage liver disease. In addition, we assume that the possibility of introducing a urease inhibitor into the treatment plan earlier is worth further investigation.

7. Why is ammonium measured in blood drawn sublingually? Give a reference and justify why this method is chosen. Also, daily anaesthesia with isoflurane could affect the results, this should be discussed.

Based on our experience, blood collected sublingually tends to be less contaminated with ammonia as opposed to blood collected from the tail which has a higher risk of being contaminated with ammonia from urine and feces. Moreover, sublingual blood sampling for ammonia measurements was successfully utilized in previous studies performed by the group of Dr. Christina Cudalbu.^{3,10,11} The justification of the chosen method and corresponding references are now provided in the Methods section 4.13 (p. 28, lines 645-646).

In our studies with the BDL model, rats were not anaesthetised daily. The anaesthesia was used during the blood sampling for a maximum of 5 min on day 0 before the BDL surgery and days 14, 35, 37 and 39 post-surgery as well as during the *in vivo* MRS experiments at the end of the treatment on day 39 post-surgery. Moreover, it was previously shown that recurrent exposure to isoflurane anaesthesia leads to minimal effects on the brain metabolism of Wistar rats.¹² The respective reference is now provided in the Methods section (p. 28, line 650).

8. In the HE model of chronic liver failure due to biliary ligation, the treatment time may be too late. In this model, the rats 5 weeks after surgery are severely affected and begin to die. It would be better to try the treatment a week or two before. The decrease in hyperammonemia in these conditions seems very slight, and, although significant in some cases, little relevant.

The prime objective of this study was to evaluate whether 2-octynoHA was able to reduce ammonemia in cirrhotic rats. Thus, to ensure that rats were hyperammonemic prior to the first treatment administration, we waited for the liver disease and thereby hyperammonemia to progress. Therefore, the treatment was started only 5 weeks post-surgery. However, the lifespan of BDL rats is on average limited to 6-8 weeks¹³ meaning that 5 weeks post-surgery rats likely experience late-stage liver cirrhosis and multiple organ dysfunction. Therefore, the reviewer is correct in assuming that there might be limitations of our timeline for the treatment. For instance, even if ammonia production *via* urea hydrolysis is inhibited, the systemic ammonia levels might not be affected due to extremely reduced liver capacity to metabolize ammonia originating from other sources than urease activity.

We agree with the reviewer that future research on this compound should include studies in BDL rats with a longer treatment initiated 2-4 weeks post-surgery. This idea has been added in the main text (p.17, lines 379-380).

9. The effects of treatments, or lack of effect, on bilirubin have not been discussed.

Bilirubin is a sign of chronic liver disease, while the treatment targets urease in the gut and thereby aims at lowering circulating ammonia levels instead of restoring liver function. Therefore, bilirubin levels were not affected by the treatment. In fact, bilirubin levels were increasing over the course of liver disease progression and were comparable during the treatment in all groups. The corresponding discussion has been included in the main text of the manuscript (p. 13, lines 285-288).

10. A relevant problem is the lack of controls, sham-operated rats, to observe the alterations present in BDL rats, mainly in the results of metabolite levels in the cerebellum.

The group of Dr. Cristina Cudalbu has previously published brain metabolites levels in the cerebellum of untreated BDL rats.^{3,11} Moreover, results of a comparison of brain metabolites levels in untreated BDL rats vs. BDL rats treated with 2-octynoHA solution or capsule have been presented at 2023 ISMRM & ISMRT Annual Meeting & Exhibition (June 3-8, 2023; Toronto, Canada).¹⁴ In addition, a manuscript comparing three brain regions (hippocampus, striatum and cerebellum) in BDL and sham-operated rats is currently in preparation. The preliminary results have been presented at the 18th ISHEN Symposium (September 12-15, 2019; Williamsburg, Virginia)¹⁵ and the 20th ISHEN Symposium (September 27-29, 2023; Bad Zwischenahn, Germany).¹⁶

11. Why has the cerebellum been chosen to analyze changes in brain metabolites? Authors must justify it and comment that the changes can be different in different brain regions.

The brain metabolites were analyzed in the cerebellum because the results of more than 10 years of research by the group of Dr. Cristina Cudalbu on the neurometabolic profile of BDL rats indicate the cerebellum as a region displaying the most notable metabolic changes (i.e. strongest brain Gln increase) measured by ¹H MRS (manuscript is in preparation). In fact, two recent manuscripts published by Dr. Cristina Cudalbu demonstrated that a Gln increase in the cerebellum of BDL rats is more evident than that in the hippocampus.^{3,11} The corresponding justification has been added to the manuscript as well as the notion of differences between brain regions (p. 14, lines 297-301).

12. The fact that all metabolites are decreased by treatments, specially solution, seems to be a general effect, rather than specific to particular metabolites, when they are as different as glucose or the GABA neurotransmitter.

It is correct that changes in the metabolite levels of BDL rats were more evident after the treatment with 2-octynoHA solution rather than other formulations. However, it would be imprecise to say that all the metabolites decreased by the treatments. In fact, in the 2-octynoHA solution-treated group, we observed a significant decrease in Gln, Tau, Cr, PCr, tCr, Glu, GABA, glucose (Glc), N-acetylaspartate (NAA) (Fig. 5, Supplementary Fig. 18), while no changes in Ins, tCh, ascorbate (Asc), lactate (Lac), N-acetylaspartylglutamate (NAAG) and phosphoethanolamine (PE). In the group treated with 2-octynoHA capsule, fewer metabolites seemed to be affected. A significant decrease was found only for Cr, tCr, Glu, Glc and NAA vs. negative control (Fig. 5, Supplementary

Fig. 17). As for 2-octynoHA suspension, only NAA significantly decreased compared to the negative control group, whereas the rest of the metabolites were comparable. Unfortunately, at this stage, it is not possible to conclude on the origin of a decrease in some metabolites and their role in the pathophysiology of HE. Perhaps, the combination of the longitudinal ¹H MRS experiments allowing comparison of metabolites before and after the treatment together with the behavioral tests might gain a better understanding of the role of the treatment in these changes as well as their relevance. The corresponding discussion was added in the main text (p. 17, lines 385-387).

In summary, the work deals with a complete study on the optimization of the best urease inhibitor and the best formulation to be able to use it as a treatment to reduce hyperammonemia, which could improve HE. In addition, the gut absorption, pharmacokinetics, stability and effect of its main metabolite are evaluated. However, the results in the *in vivo* models used are not very conclusive and it does not seem that the chosen inhibitor has very significant effects on hyperammonemia, especially in the chronic liver failure model, compared to other treatments previously used to reduce hyperammonemia in HE. This should be better discussed and the study expanded with better models.

Although rifaximin is used in the management of HE to reduce hyperammonemia, recent studies on the efficacy of rifaximin in the chronic liver disease model (BDL rats) did not demonstrate any efficacy of the antibiotic in terms of lowering ammonemia as well as brain Gln.^{3,4} In our *in vivo* studies in BDL rats, hyperammonemia was improved after 3 days of treatment with 2-octynoHA solution and this was followed by the decrease in brain Gln levels after 5 days of treatment supporting the therapeutic potential of 2-octynoHA. The transient effect on ammonemia in both BDL and DEN-induced liver damage models might be attributed to the severe deterioration of liver function at the end of the study leading to decreased capacity of the liver to metabolize ammonia. Thus, in the future, the sustained efficacy of a urease inhibitor can be assessed in the studies with prolong treatment initiated at an earlier stage. Furthermore, the efficacy of 2-octynoHA treatment might be improved through optimization of formulation ensuring colonic delivery as well as increased solubility and stability. The corresponding discussion has been expanded in the main text (p. 17, lines 372-394).

References

1. Shen, S. & Kozikowski, A. P. Why hydroxamates may not be the best histone deacetylase inhibitors—what some may have forgotten or would rather forget? *ChemMedChem* **11**, 15–21 (2016).
2. Badawy, S. I. F. & Hussain, M. A. Microenvironmental pH modulation in solid dosage forms. *Journal of Pharmaceutical Sciences* **96**, 948–959 (2007).
3. Flatt, E. *et al.* Probiotics combined with rifaximin influence the neurometabolic changes in a rat model of type C HE. *Sci Rep* **11**, 17988 (2021).
4. Thabut, D. *et al.* Sodium benzoate and rifaximin are able to restore blood-brain barrier integrity in the cirrhotic rats. *Intensive Care Medicine Experimental* **3**, A691 (2015).
5. Gómez-Lado, N. *et al.* Gastrointestinal Tracking and Gastric Emptying of Coated Capsules in Rats with or without Sedation Using CT imaging. *Pharmaceutics* **12**, 81 (2020).
6. Langerholm, T., Maragkoudakis, P. A., Wollgast, J., Gradisnik, L. & Cencic, A. Novel and established intestinal cell line models – An indispensable tool in food science and nutrition. *Trends Food Sci Technol* **22**, S11–S20 (2011).
7. Tolba, R., Kraus, T., Liedtke, C., Schwarz, M. & Weiskirchen, R. Diethylnitrosamine (DEN)-induced carcinogenic liver injury in mice. *Lab Anim* **49**, 59–69 (2015).
8. Mukherjee, D. & Ahmad, R. Dose-dependent effect of N'-Nitrosodiethylamine on hepatic architecture, RBC rheology and polypeptide repertoire in Wistar rats. *Interdiscip Toxicol* **8**, 1–7 (2015).
9. Reebye, V. *et al.* Gene activation of CEBPA using saRNA: preclinical studies of the first in human saRNA drug candidate for liver cancer. *Oncogene* **37**, 3216–3228 (2018).
10. Račkayová, V. *et al.* Probiotics improve the neurometabolic profile of rats with chronic cholestatic liver disease. *Sci Rep* **11**, 2269 (2021).
11. Mosso, J. *et al.* PET CMRglc mapping and 1H-MRS show altered glucose uptake and neurometabolic profiles in BDL rats. *Analytical Biochemistry* **647**, 114606 (2022).
12. Račkayová, V. *et al.* Late post-natal neurometabolic development in healthy male rats using 1 H and 31 P magnetic resonance spectroscopy. *J Neurochem* **157**, 508–519 (2021).
13. DeMorrow, S., Cudalbu, C., Davies, N., Jayakumar, A. R. & Rose, C. F. 2021 ISHEN guidelines on animal models of hepatic encephalopathy. *Liver International* **41**, 1474–1488 (2021).
14. Simicic, D. *et al.* Novel urease inhibitor as potential treatment of hepatic encephalopathy led to brain glutamine decrease (abstract 3865). in *2023 ISMRM & ISMRT Annual Meeting & Exhibition* (2023).
15. Simicic, D. *et al.* In vivo longitudinal 1H MRS study of hippocampal, cerebral and striatal metabolic changes in the adult brain using an animal model of chronic hepatic encephalopathy (abstract 33). *Official journal of the American College of Gastroenterology | ACG* **114**, S17 (2019).
16. Simicic, D. *et al.* Brain regional differences in the developing brain using an animal model of type C hepatic encephalopathy (abstract 36). *Official journal of the American College of Gastroenterology | ACG* **118**, S24 (2023).

REVIEWER COMMENTS

Reviewer #1 (Remarks to the Author):

With the same premise that I used in my reviewing of the original manuscript, namely that "my competence does not cover the cytotoxicity nor the pharmacokinetic sections of the study, nor the in vivo assays, so my comments will be limited to the chemistry and structural studies", I can see the authors have simply removed the section of the manuscript that I asked to improve with my comment about "the failure to report and comment on the existence of four crystal structures of different ureases inhibited by hydroxamates (PDB codes 4H9M, 4UBP, 1E9Y and 1FWE). If the authors had focused only on the pharmaceutical aspects of the proposed new drug, it would have been more acceptable. However, the fact that they tried to understand the binding mechanism of 2-octynoHA using molecular dynamics (paragraph 2.3) without ever citing the known structures of urease-hydroxamate, reveals that the authors have not done their homework" I do not find that the authors have made any effort to "improve" this section: they could have removed the useless and misleading section of molecular dynamics and still acknowledge previous biochemical and structural studies of urease inhibition by hydroxamates. The authors simply gave up on discussing the structural aspects of their study, which would have been useful as describing the state of the art for this class of inhibitors. Moreover, they also easily dismissed my second comment concerning the stability of their proposed inhibitor, responding with wishful comments about the possibility, which they have not tested in vivo, to use acidic excipients to increase the lifetime of the potential drug, while admitting themselves that "the currently identified lead structure may not be stable enough in clinical trials and that its structure would have to be tuned to improve its half-life". For these reasons, I do not find the revised manuscript significant enough to grant publication in Nat. Commun., and I must therefore recommend rejection.

Reviewer #2 (Remarks to the Author):

I would like to thank the authors for providing answers, clarification and significant changes to their manuscript that make it acceptable for publication.

Reviewer #3 (Remarks to the Author):

The authors have responded satisfactorily to my comments and have modified the manuscript accordingly, thus clarifying the significance and conclusions of their work.

I have no further comments, and my recommendation is that the manuscript can be accepted.

Reviewer #1:

With the same premise that I used in my reviewing of the original manuscript, namely that "my competence does not cover the cytotoxicity nor the pharmacokinetic sections of the study, nor the in vivo assays, so my comments will be limited to the chemistry and structural studies", I can see the authors have simply removed the section of the manuscript that I asked to improve with my comment about "the failure to report and comment on the existence of four crystal structures of different ureases inhibited by hydroxamates (PDB codes 4H9M, 4UBP, 1E9Y and 1FWE). If the authors had focused only on the pharmaceutical aspects of the proposed new drug, it would have been more acceptable. However, the fact that they tried to understand the binding mechanism of 2-octynoHA using molecular dynamics (paragraph 2.3) without ever citing the known structures of urease-hydroxamate, reveals that the authors have not done their homework" I do not find that the authors have made any effort to "improve" this section: they could have removed the useless and misleading section of molecular dynamics and still acknowledge previous biochemical and structural studies of urease inhibition by hydroxamates. The authors simply gave up on discussing the structural aspects of their study, which would have been useful as describing the state of the art for this class of inhibitors.

We thank the reviewer for emphasizing the importance of acknowledging previous biochemical and structural studies of urease inhibition by hydroxamate. We agree that the description of the active site of ureases as well as the inhibition mechanism and binding pose for hydroxamic acids should be mentioned in the work dedicated to exploration of novel hydroxamate-based urease inhibitors. Therefore, this information has been added to the manuscript (pages 4-5, lines 66-96). Additionally, as recommended, we acknowledged the known crystal structures of urease-hydroxamate (page 5, line 80-96) as they contributed to our better understanding of the binding pose.

Furthermore, we would like to comment on removing the section of molecular dynamics (MD) in our revised manuscript. We recognize that in the first iteration of the manuscript the outcomes from MD simulations were not validated against the experimental binding pose. Moreover, there was also an inaccurate statement, since we employed structure PDB ID: 1E9Y^{1,2} (instead of PDB ID: 1E9Z^{2,3}) as the starting point, which includes acetohydroxamic acid (AHA). In order to address the reviewer's comments regarding the model validation from the previous review, we revised the modeling activity from the beginning. We found out that previously employed bonded model introduced some artifacts and led to an incorrect binding pose. The bonded model was initially employed to preserve the structural integrity of the binding site, since nickel ions were released in the surrounding solution with a non-bonded model. Such issue could arise from the assignment of the protonation states by the H++ server,⁴ and, specifically, by the choice of the internal dielectric constant. In the initially submitted MD experiments, the internal dielectric constant was equal to 10. However, during the revision process, we performed additional simulations and increased the value of the internal dielectric constant to 20 (which is within the range of the recommended values for residues closer to the protein's surface) which allowed us to preserve the structure of the binding site with a simpler non-bonded model. We repeated the unbiased MD simulations for the same ligands (i.e. AHA, HHA, OHA, NHA, DHA, LHA, 2-octenoHA, 2-octynoHA, and 7-octynoHA) considered in the initially submitted manuscript, and the binding pose of AHA showed a better agreement with the experimental binding pose, thus confirming the bias

introduced by the bonded model. The post-processing using Molecular Mechanics Generalized Born Surface Area (MMGBSA)⁵ resulted in the analogous outcomes; specifically, the interaction energy was not able to recapitulate the data from the *in vitro* studies. Since the simulation data only represented a minor fraction of the paper and did not affect the main conclusion of the paper, we decided to remove them and eventually address this issue at a later stage.

Moreover, they also easily dismissed my second comment concerning the stability of their proposed inhibitor, responding with wishful comments about the possibility, which they have not tested in vivo, to use acidic excipients to increase the lifetime of the potential drug, while admitting themselves that "the currently identified lead structure may not be stable enough in clinical trials and that its structure would have to be tuned to improve its half-life". For these reasons, I do not find the revised manuscript significant enough to grant publication in Nat. Commun., and I must therefore recommend rejection.

We would first like to mention that the current compound showed a greater ammonia lowering effect than the reference drug. We also believe that the activity could potentially be further improved by modifying certain formulation parameters. Our assumption on using acidifying excipients to stabilize 2-octynoHA was based on the data acquired from the *in vitro* stability tests. We showed that the degradation rate of 2-octynoHA was slower at pH 6.3 than at pH 6.8 (**Fig. 2e**) which allowed us to hypothesize that formulation containing acidifying excipients might be a potential approach to stabilize 2-octynoHA and prolong its activity. We recognize the importance of exploring this approach *in vivo*, however, conducting such studies in a disease model would require substantial efforts.

In our studies in BDL rats, for the colon-targeted delivery, capsules were coated with pH-sensitive polymer, Eudragit L100, which dissolves at pH above 6. Since dissolution of the coating is determined by pH, additional acidifying excipients in the formulation might affect capsule disintegration rate leading to delayed disintegration or failure to disintegrate. Therefore, to evaluate whether a new formulation with acidifying excipients is able to prolong activity of 2-octynoHA and improve its efficacy *in vivo*, the capsule coating will have to be first completely reengineered. Thus, addressing the reviewer's concern would involve a complex set of *in vitro* tests (i.e. screening of coating materials and the layer thickness, assessment of capsule disintegration in gastrointestinal conditions) as well as *in vivo* experiments. This work might be performed in the next steps of the project.

References

1. Ha, N.-C., Oh, S.-T. & Oh, B.-H. Crystal structure of *Helicobacter pylori* urease in complex with acetohydroxamic acid. <https://doi.org/10.2210/pdb1e9y/pdb> (2001).
2. Ha, N.-C. *et al.* Supramolecular assembly and acid resistance of *Helicobacter pylori* urease. *Nat Struct Mol Biol* **8**, 505–509 (2001).
3. Ha, N.-C., Oh, S.-T. & Oh, B.-H. Crystal structure of *Helicobacter pylori* urease. <http://dx.doi.org/10.2210/pdb1e9z/pdb> (2001).
4. Anandkrishnan, R., Aguilar, B. & Onufriev, A. V. H++ 3.0: automating pK prediction and the preparation of biomolecular structures for atomistic molecular modeling and simulations. *Nucleic Acids Research* **40**, W537–W541 (2012).

5. Miller, B. R. *et al.* MMPBSA.py: an Efficient program for end-state free energy calculations. *J Chem Theory Comput* **8**, 3314–3321 (2012).

REVIEWERS' COMMENTS

Reviewer #1 (Remarks to the Author):

The authors have revised their manuscript according to most of my observations. I consider the paper now acceptable for publication.